# COMMUNICATE THEN ADAPT: AN EFFECTIVE DECENTRALIZED ADAPTIVE METHOD FOR DEEP TRAINING

## ABSTRACT

Decentralized adaptive gradient methods, in which each node averages only with its neighbors, are critical to save communication and wall-clock training time in deep learning tasks. While different in concrete recursions, existing decentralized adaptive methods share the same algorithm structure: each node scales its gradient with information of the past squared gradients (which is referred to as the adaptive step) *before* or *while* it communicates with neighbors. In this paper, we identify the limitation of such *adapt-then/while-communicate* structure: it will make the developed algorithms highly sensitive to data heterogeneity, and hence deviate their limiting points from the stationary solution. To overcome this limitation, we propose an effective decentralized adaptive method with a *communicate-then-adapt* structure, in which each node conducts the adaptive step after finishing the neighborhood communications. The new method is theoretically guaranteed to converge to the stationary solution in the non-convex scenario. Experimental results on a variety of CV/NLP tasks show that the proposed algorithm has a clear superiority to other existing decentralized adaptive methods.

## 1 INTRODUCTION

Decentralized SGD (Lopes & Sayed, 2008; Nedic & Ozdaglar, 2009; Chen & Sayed, 2012; Lian et al., 2017; Assran et al., 2019) is an emerging training approach for deep learning known for its much less communication overhead. In contrast to parallel SGD in which a global averaging across all computing nodes is required per iteration, decentralized SGD does not involve any global operations. Building upon partial averaging, in which each node only needs to compute the locally averaged model within its neighborhood, decentralized SGD can save remarkable communications and training time in large-scale distributed deep learning tasks compared to parallel SGD.

Although simple to use, the vanilla decentralized SGD sometimes suffers from the slow convergence. Inspired by the well-documented success of adaptive methods such as AdaGrad (Duchi et al., 2011a), Adam (Kingma & Ba, 2014) and AMSGrad (Reddi et al., 2019), several decentralized adaptive methods (Nazari et al., 2019; Lin et al., 2021) have been proposed to accelerate decentralized SGD training. While these algorithms have achieved remarkable success in several practical applications, they have also been observed to not converge to the desired solution (i.e., global optimal solution in the convex scenario or stationary solution in the non-convex scenario) in some other settings. For example, it has been observed in the convex setting (see Sec. 3) that DAdam (Nazari et al., 2019) and QG-DAdam (Lin et al., 2021) do not converge to the global optimal solution.

This paper studies this situation in detail. We rigorously uncover the reason why DAdam and QG-DAdam fail to achieve the desired solution, and propose a novel decentralized adaptive method to resolve the convergence issue. In particular, we make the following key contributions:

- We find the algorithms DAdam and QG-DAdam, while different in concrete recursions, share a similar structure: each node scales its gradient with the past squared gradients (which is referred to as the adaptive step) *before* or *while* it communicates with neighbors. We identify the limitation of such *adapt-then/while-communicate* structure: it will make the developed algorithms highly sensitive to data heterogeneity, and hence deviate their limiting points from the desired solution.

- To overcome these limitations, we propose a novel *communicate-then-adapt* algorithm structure, in which each node will conduct the adaptive step after all neighborhood communications. It is *not* just trivially switching the order of communication step and adaptive step. The key component to guarantee the effectiveness of the communicate-then-adapt structure is the utilization of the

*decentralized augmented gradient* (DAG) rather than the standard stochastic gradient in algorithm development. The newly proposed algorithm, which is coined as DAG-Adam, can provably converge to the desired solution. While this paper mainly focuses on the vanilla Adam algorithm, the core idea behind DAG-Adam can be easilty extended to AMSGrad or other adaptive methods.

- Experimental results on a variety of computer vision and natural language processing tasks show that DAG-Adam outperforms various existing state-of-the-art decentralized adaptive baselines under different network and data configurations. Furthermore, the performance of our proposed algorithm is persistently competitive to the centralized counterpart.

**Related work on decentralized optimization and deep training.** Decentralized optimization was extensively studied in the control and signal processing community. The first decentralized algorithms on general optimization problems include decentralized gradient descent (Nedic & Ozdaglar, 2009), diffusion (Lopes & Sayed, 2008; Chen & Sayed, 2012; Sayed, 2014), and dual averaging (Duchi et al., 2011b). After that, various primal-dual algorithms come out to further speed up the convergence, and they are based on alternating direction method of multipliers (Adam) (Shi et al., 2014), explicit bias-correction (Shi et al., 2015; Yuan et al., 2019; Li et al., 2019), gradient tracking (Xu et al., 2015; Di Lorenzo & Scutari, 2016; Nedic et al., 2017; Qu & Li, 2018; Lu et al., 2019), and dual acceleration (Scaman et al., 2017; Uribe et al., 2020).

In deep learning tasks, decentralize SGD, which was established in (Lian et al., 2017) to achieve the same linear speedup as parallel SGD in convergence rate, has attracted a lot of attentions. Many efforts have been made to extend the algorithm to directed topologies (Assran et al., 2019), time-varying topologies (Koloskova et al., 2020), asynchronous settings (Lian et al., 2018), and data-heterogeneous scenarios (Tang et al., 2018; Xin et al., 2020; Lin et al., 2021; Yuan et al., 2021). With careful consensus control (Kong et al., 2021) or periodic global averaging (Chen et al., 2021b), decentralize SGD can achieve $1.3 \sim 2\times$ training time speedup without severe performance degradation. Techniques such as quantization/compression (Alistarh et al., 2017; Bernstein et al., 2018; Koloskova et al., 2019a;b; Tang et al., 2019; Liu et al., 2020), periodic updates (Stich, 2019; Koloskova et al., 2020; Yu et al., 2019), and lazy communication (Chen et al., 2018a; Liu et al., 2019b) were also integrated into decentralized SGD to further reduce communication overheads.

There are also a few studies on accelerated variants of decentralized SGD, and most of them are on (static) momentum acceleration. (Assran et al., 2019; Gao & Huang, 2020) propose to run a local momentum SGD step first before the partial averaging is conducted. Another work (Yu et al., 2019) imposes an additional partial averaging over momentum to increase stability. Recent works (Lin et al., 2021; Yuan et al., 2021) developed strategies that can remove the momentum-incurred bias in decentralized momentum SGD. All these methods are not with adaptive strategies to scale gradients.

**Related work on adaptive gradient method.** Adaptive gradient methods, with AdaGrad (Duchi et al., 2011a) and Adam (Kingma & Ba, 2014) as two representatives, have shown strong performance in training deep neural networks. With adaptive adjustment in the gradient direction and automatic tune in the learning rate, adaptive gradient methods can boost the performance of SGD training significantly. In spite of its remarkable empirical success, Adam suffers from a convergence issue: it may not converge to the desired solution with a fixed mini-batch size (Reddi et al., 2019; Zaheer et al., 2018). Many algorithms have been proposed to resolve this issue. AMSGrad (Reddi et al., 2019) preserves the long-term memory of past gradients to improve convergence, and (Zaheer et al., 2018) suggests the usage of increasing mini-batch sizes in convergence guarantees. (Chen et al., 2018b) has studied the convergence of a family of Adam-type algorithms. (Zhou et al., 2018) provides a better convergence rate for AMSGrad. From the empirical side, RAdam (Liu et al., 2019a) and AdamW (Loshchilov & Hutter, 2018) can significantly improve Adam performance.

The exploration in decentralized adaptive methods are rather limited. DAdam (Nazari et al., 2019) is the first consensus-based adaptive methods for distributed optimization to our knowledge. (Lin et al., 2021) proposes QG-DAdam, which utilizes quasi-global momentum to locally approximates the globally descent direction. A recent work (Chen et al., 2021a) proposes a unified framework that incorporates various adaptive methods into decentralized setting. While these methods have shown strong empirical performance in several practical applications, they either suffer from unstable convergence or heavy communications. We leave more discussion on their limitations in Sec. 2. Adaptive gradient methods have also been extended to the federated learning setting in which multiple clients cooperate to learn a model under the supervision of a central server (McMahan et al., 2017). Useful references in this direction can be found in (Xie et al., 2019; Reddi et al., 2020).

## 2  ADAPT-THEN/WHILE-COMMUNICATE STRUCTURE AND ITS LIMITATION

**Problem.** Suppose $n$ computing nodes cooperate to solve the distributed optimization problem:

$$\min_{x \in \mathbb{R}^d} f(x) = \frac{1}{n} \sum_{i=1}^{n} f_i(x), \quad \text{where} \quad f_i(x) := \mathbb{E}_{\xi_i \sim D_i} F(x; \xi_i) \tag{1}$$

In the above problem, $f_i(x)$ is local to node $i$, and random variable $\xi_i$ denotes the local data that follows distribution $D_i$. We do *not* assume each distribution $D_i$ is the same across all nodes.

**Network topology and weights.** We assume all computing nodes are connected by a network topology. We define $w_{ij}$, the weight to scale information flowing from node $j$ to node $i$, as follows:

$$w_{ij} \begin{cases} > 0 & \text{if node } j \text{ is connected to } i, \text{ or } i = j; \\ = 0 & \text{otherwise.} \end{cases} \tag{2}$$

$\mathcal{N}_i := \{j | w_{ij} > 0\}$ is defined as the set of neighbors of node $i$ which also includes node $i$ itself and the *weight matrix* $W := [w_{ij}]_{i,j=1}^{n} \in \mathbb{R}^{n \times n}$ are denoted as a matrix that stacks the weights of all nodes. This matrix $W$ characterizes the sparsity and connectivity of the underlying topology.

**Partial averaging.** Decentralized methods are based on *partial averaging* within neighborhood defined by the network topology. With weights $\{w_{ij}\}$ and the set of neighbors $\mathcal{N}_i$, the partial averaging operation of node $i$ can be expressed as

$$\text{Partial averaging:} \quad x_i^+ \leftarrow \sum_{j \in \mathcal{N}_i} w_{ij} x_j. \tag{3}$$

Partial averaging requires much less communication than global averaging on sparse topologies.

**Assumptions.** We will make the following standard assumptions throughout the paper:

**A.1** [SMOOTHNESS] Each $f_i(x)$ is $L$-smooth, i.e., $\|\nabla f_i(x) - \nabla f_i(y)\| \leq L\|x - y\|$ for any $x, y \in \mathbb{R}^d$.

**A.2** [GRADIENT NOISE] The random sample $\xi_i^{(t)}$ is independent of each other for any $k$ and $i$. We also assume $\mathbb{E}[\nabla F(x; \xi_i)] = \nabla f_i(x)$ and $\mathbb{E}\|\nabla F(x; \xi_i) - \nabla f_i(x)\|^2 \leq \sigma^2$.

**A.3** [WEIGHT MATRIX] The weight matrix $W$ is symmetric and doubly-stochastic, i.e. $W\mathbb{1} = \mathbb{1}$ and $\mathbb{1}^T W = \mathbb{1}^T$, where $\mathbb{1}$ is an all-ones vector. Further, we assume $\|W - \frac{1}{n}\mathbb{1}\mathbb{1}^T\|_2 \leq \rho \in (0, 1)$.

**A.4** [BOUNDED GRADIENT] The loss function $F$ has bounded gradient, i.e., the maximum norm $\|\nabla F(x; \xi_i)\|_\infty \leq G$ for all $x$ and $\xi_i$, and $i \in [n]$.

**Notation.** Given a constant $n$, we let $[n] := \{1, \cdots, n\}$. Given a vector $x \in \mathbb{R}^d$, we let $\text{diag}(x) = \text{diag}\{x_1, \cdots, x_d\} \in \mathbb{R}^{d \times d}$ be a diagonal matrix. Given two vectors $x \in \mathbb{R}^d$ and $y \in \mathbb{R}^d$, we let $x \odot y$ be the element-wise product between $x$ and $y$.

### 2.1  THE ADAPT-WHILE-COMMUNICATE STRUCTURE AND ITS LIMITATION

**Adapt-While-Communicate Structure.** Algorithm 2 in Appendix B lists recursions for DAdam (Nazari et al., 2019). The critical step in DAdam, which is highlighted in Algorithm 2, is as follows

$$x_i^{(t+1)} = \overbrace{\sum_{j \in \mathcal{N}_i} w_{ij} x_j^{(t)}}^{\text{communicate}} - \overbrace{\frac{\gamma \, m_i^{(t)}}{\sqrt{\hat{v}_i^{(t)} + \epsilon}}}^{\text{adapt}}, \quad \forall \, i \in [n]. \tag{4}$$

where $m_i^{(t)}$ is the momentum, $\hat{v}_i^{(t)}$ is an estimate of $\nabla F(x_i^{(t)}; \xi_i^{(t)}) \odot \nabla F(x_i^{(t)}; \xi_i^{(t)})$, and $\epsilon$ is a small positive constant. Their updates can be referred to Algorithm 2. In recursion (4), it is observed that node $i$ in DAdam conducts the adaptive step (i.e., scales the gradient by the square root of exponential moving averages of past squared gradients) *while* it communicates with neighbors. We call such structure utilized in DAdam as *adapt-while-communicate*.

**Fixed point condition.** Now we examine the fixed point achieved by DAdam and verify whether it satisfies the optimal condition to problem (1). To this end, we let $x_i^{(t)} \to x_i^\infty$ as $t \to \infty$ for $i \in [n]$. To remove the the influence of gradient noise, we assume each node $i$ can access the full-batch gradient $\nabla f_i(x)$. If we define $H_i := \text{diag}(\sqrt{\nabla f_i(x_i^\infty) \odot \nabla f_i(x_i^\infty)})$ to be a diagonal matrix associated with node $i$, the fixed point condition for DAdam is (see derivation in Appendix B.2):

$$x_i^\infty = \sum_{j \in \mathcal{N}_i} w_{ij} x_j^\infty - \gamma(H_i + \epsilon I)^{-1} \nabla f_i(x_i^\infty), \quad \forall\, i \in [n]. \tag{5}$$

$$\implies \quad \frac{1}{n} \sum_{i=1}^n (H_i + \epsilon I)^{-1} \nabla f_i(x^\infty) = 0 \quad (\text{if } x_1^\infty = \cdots = x_n^\infty = x^\infty \text{ in the limit}) \tag{6}$$

**Fixed point in DAdam is deviated from the desired solution.** In the data-heterogeneous scenario where data sample $\xi_i$ follows different distribution $D_i$, it holds that $f_i(x) \neq f_j(x)$. If we further assume $\nabla f_i(x) \neq \nabla f_j(x)$, it follows that $H_i \neq H_j$, for any $i \neq j$. With this fact, we conclude that DAdam fixed point satisfying condition (6) does not necessarily satisfy the optimality condition for problem (1), i.e., $\frac{1}{n} \sum_{i=1}^n \nabla f_i(x^\infty) = 0$. This implies the fixed point of DAdam is *not* the stationary solution to problem (1). In particular, the following proposition derives the distance between the DAdam fixed point and global optimal solution to problem (1) when the cost function is strongly-convex (Proof is in Appendix B.3).

**Proposition 1** *Under Assumptions A.1, A.3 and A.4, if each $f_i(x)$ is further assumed to be $\mu$-strongly-convex, and the full-batch gradient $\nabla f_i(x)$ is accessed per iteration, then DAdam (Algorithm 2) cannot converge to $x^\star$, i.e., the global solution to problem (1). In particular, the distance between DAdam limiting point and $x^\star$ can be characterized as follows:*

$$\lim_{t \to \infty} \sum_{i=1}^n \|x_i^{(t)} - x^\star\|^2 = O\Big(\frac{\gamma^2 G^2}{(1-\rho)^2 \epsilon^2} + B^2\Big) \tag{7}$$

*where $B := \|\frac{1}{n} \sum_{i=1}^n (H_i + \epsilon I)^{-1} \nabla f_i(x^\infty) - \frac{1}{n} \sum_{i=1}^n \nabla f_i(x^\infty)\| \neq 0$ is a constant bias term. Furthermore, the $B^2$ term in (7) is necessary and cannot be removed by an improved analysis; there exists strongly convex and smooth functions $f_i(x)$ such that $\lim_{t \to \infty} \sum_{i=1}^n \|x_i^{(t)} - x^\star\|^2 = \Omega(B^2)$.*

It is observed in Proposition 1 that the distance between the DAdam fixed point and the optimal solution cannot diminish to zero as learning rate $\gamma \to 0$. Such non-vanishing limiting bias $B^2$ is caused by the adapt-while-communicate structure in DAdam. It is worth noting that a recent work (Chen et al., 2021a) has pointed out that DAdam cannot converge to the stationary solution by constructing a counter-example. However, (Chen et al., 2021a) does not clarify the reason why DAdam fails to converge to the stationary solution, and how far it can be from the desired solution.

## 2.2 THE ADAPT-THEN-COMMUNICATE STRUCTURE AND ITS LIMITATION

**Adapt-Then-Communicate Structure.** Algorithm 3 in Appendix B lists the recursions for QG-DAdam (Lin et al., 2021). The critical step, which is highlighted in Algorithm 3, is as follows

$$x_i^{(t+1)} = \sum_{j \in \mathcal{N}_i} w_{ij} \Big(x_j^{(t)} - \gamma \frac{m_j^{(t)}}{\sqrt{v_j^{(t)}} + \epsilon}\Big) \tag{8}$$

It is observed that node $i$ in the above recursion first conducts the adaptive step, and *then* it communicates with neighbors. We call the structure utilized in QG-DAdam as *adapt-then-communicate*.

**Fixed point condition and limitation.** Following the settings and arguments in Sec. 2.1, we can derive the fixed point condition for QG-DAdam as follows (see the derivation in Appendix B.5):

$$x_i^\infty = \sum_{j \in \mathcal{N}_i} w_{ij} \Big(x_j^\infty - \gamma \beta_1 (\sqrt{\beta_2} H_j + \epsilon I)^{-1} \nabla f_j(x_j^\infty)\Big), \quad \forall\, i \in [n]. \tag{9}$$

$$\implies \quad \frac{1}{n} \sum_{i=1}^n (\sqrt{\beta_2} H_i + \epsilon I)^{-1} \nabla f_i(x^\infty) = 0 \quad (\text{if } x_1^\infty = \cdots = x_n^\infty = x^\infty \text{ in the limit}) \tag{10}$$

where $\beta_1, \beta_2 \in (0, 1)$ are momentum parameters listed in Algorithm 3. Similar to DAdam, the above condition implies that the fixed point of QG-DAdam is also deviated from the stationary solution. In particular, the distance between QG-DAdam fixed point and the global solution $x^\star$ can be characterized for the strongly-convex scenario as follows (proof is in Appendix B.6):

**Proposition 2** *Under the same conditions as in Proposition 1, the distance between QG-DAdam fixed point and the global solution $x^\star$ can be characterized as follows:*

$$\lim_{t \to \infty} \sum_{i=1}^{n} \|x_i^{(t)} - x^\star\|^2 = O\Big(\frac{\gamma^2 G^2}{(1-\rho)^2 \epsilon^2} + B^2\Big) \tag{11}$$

*where $B := \|\frac{1}{n} \sum_{i=1}^{n} (\sqrt{1-\beta_2} H_i + \epsilon I)^{-1} \nabla f_i(x_i^\infty) - \frac{1}{n} \sum_{i=1}^{n} \nabla f_i(x_i^\infty)\| \neq 0$ is a constant bias. Furthermore, the $B^2$ term in (7) is necessary and cannot be removed by an improved analysis; there exists strongly convex and smooth functions $f_i(x)$ such that $\lim_{t \to \infty} \sum_{i=1}^{n} \|x_i^{(t)} - x^\star\|^2 = \Omega(B^2)$.*

Similar to DAdam, it is observed that the distance between the QG-DAdam fixed point and the optimal solution cannot diminish to zero as learning rate $\gamma \to 0$.

### 2.3 GT-DADAM: A DECENTRALIZED ADAPTIVE METHOD BASED ON GRADIENT TRACKING

It is observed from the fixed-point condition (6) that DAdam cannot converge to the stationary solution due to the discrepancy between $H_i$'s. A recent work (Chen et al., 2021a) proposes a gradient tracking based DAdam that enables each local different $H_i$ to converge to $\bar{H} = \frac{1}{n} \sum_{i=1}^{n} H_i$ in theory, and therefore, its limiting fixed point satisfies $\frac{1}{n} \sum_{i=1}^{n} \nabla f_i(x^\infty) = 0$, the optimality condition to problem (1). GT-DAdam also utilizes the adapt-while-communicate structure. While it can converge to the stationary solution to problem (1) as learning rate $\gamma \to 0$, it incurs *twice* amount of the communication overhead as DAdam. The distance between GT-DAdam fixed point and the global solution $x^\star$ can be characterized in Proposition 4 in Appendix B.7 for the strongly-convex scenario. In the convex scenario, the distance between GT-DAdam fixed point and the global solution can be worse than that of the proposed DAG-Adam, see Table 5 in Appendix C.3 and Fig.1 in Sec. 3.

## 3 DECENTRALIZED AUGMENTED GRADIENT ADAM

Now that we understand that the adapt-while/then-communicate structure in decentralized adaptive methods either deviates the limiting point from the desired solution (e.g., DAdam and QG-DAdam), or incurs more expensive communication overhead (GT-DAdam), this section will propose a new algorithm that can overcome the limitation of the adapt-while/then-communicate structure.

**Decentralized SGD interprets as standard SGD.** Without loss of generality, we assume each node $i$ can access the accurate gradient $\nabla f_i(x)$ in decentralized SGD recursions. In this scenario, decentralized SGD, according to (Yuan et al., 2016), can be written as

$$x_i^{(t+1)} = \sum_{j \in \mathcal{N}_i} w_{ij} x_j^{(t)} - \gamma \nabla f_i(x_i^{(t)}) \tag{12}$$

$$= x_i^{(t)} - \gamma \big( \underbrace{\nabla f_i(x_i^{(t)}) + \frac{1}{\gamma}(x_i^{(t)} - \sum_{j \in \mathcal{N}_i} w_{ij} x_j^{(t)})}_{\text{augmented gradient}} \big), \quad \forall i \in [n]. \tag{13}$$

In other words, decentralized SGD (12) can be regarded as a standard SGD recursion (13) to solve

$$\min_{\{x_i\}_{i=1}^n} \quad \mathcal{L}(\{x_i\}_{i=1}^n) := \sum_{i=1}^{n} f_i(x_i) + \frac{1}{2\gamma}\Big(\sum_{i=1}^{n} \|x_i\|^2 - \sum_{i=1}^{n}\sum_{j=1}^{n} w_{ij} x_i x_j\Big). \tag{14}$$

We call above as an augmented loss function, which is an approximate penalty problem to the target problem (1). The term $\sum_{i=1}^{n} \|x_i\|^2 - \sum_{i,j} w_{ij} x_i x_j$ is convex when the weight matrix $W = [w_{ij}]$ is symmetric and doubly-stochastic, and it equals 0 if $x_1 = \cdots = x_n$. Apparently, it is a regularization term that promotes consensus among all nodes.

**Decentralized Augmented Gradient Adam (DAG-Adam).** Inspired by the fact that decentralized SGD can be interpreted as a standard SGD with an augmented gradient (13), we can easily integrate the augmented gradient to the standard Adam method (Kingma & Ba, 2014). Decentralized Augmented Gradient Adam, or DAG-Adam for short is listed in Algorithm 1. It is observed that all communications occur when the stochastic augmented gradient is computed (see the highlighted part in Algorithm 1). After that, each node $i$ will conduct an adaptive step, which is untangled with communication. We call the structure utilized in DAG-Adam as *communicate-then-adapt*. This adjustment of the algorithm structure is crucial. The stochastic gradient that used to compute $m_i^{(t)}$ and

---

**Algorithm 1** DAG-Adam

---

**Initialize** $x_i^{(0)}$ arbitrarily; let $\beta_1, \beta_2, \in [0,1)$; let $m_i^{(0)} = v_i^{(0)} = 0$; set $\gamma, \nu$ properly;
**For** $t = 0, 1, 2, ..., T-1$, every node $i$ **do in parallel**

Sample $\hat{g}_i^{(t)} = \nabla F(x_i^{(t)}; \xi_i^{(t)}) \boxed{+ \frac{\nu}{\gamma}\Big(x_i^{(t)} - \sum_{j \in \mathcal{N}_i} w_{ij} x_j^{(t)}\Big)}$     ▷ stochastic augmented gradient;

$m_i^{(t)} = \beta_1 m_i^{(t-1)} + (1-\beta_1)\hat{g}_i^{(t)};$
$v_i^{(t)} = \beta_2 v_i^{(t-1)} + (1-\beta_2)\hat{g}_i^{(t)} \odot \hat{g}_i^{(t)};$
$x_i^{(t+1)} = x_i^{(t)} - \gamma \frac{m_i^{(t)}}{\sqrt{v_i^{(t)}} + \epsilon}$     ▷ adaptive step;

---

$v_i^{(t)}$ have incorporated the neighbor's information, which will correct the deviation in fixed point that introduced by different $v_i$ (and $H_i$).

Note that a parameter $\nu > 0$ is added to the adjust the priority to gradient descent, or to promote consensus. Empirically, we can set $\nu = 1$ for the convex scenario. But for the non-convex deep learning problems, the choice of $\nu$ will heavily influence both convergence rate and performance. We extensively discuss the influence of $\nu$ in Appendix D and empirically examine it in the later experiment section. While this paper we only focus on the extension of the vanilla Adam method, the core ideas such as the augmented gradient and the communicate-then-adapt structure can be extended to AMSGrad or other adaptive gradient methods.

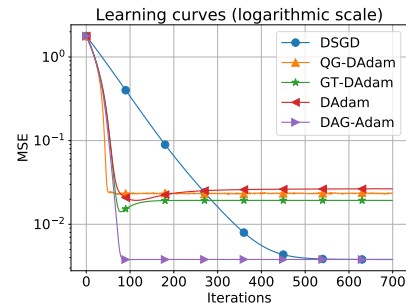

Figure 1: Convergence comparison between various algorithms for a full-batch logistic regression problem.

**Fixed point condition.** The stochastic augmented gradient and communicate-then-adapt structure utilized in DAG-Adam can remove the influence of the discrepancy in $v_i^\infty$ and hence $H_i$. Following the settings and arguments in Sec. 2.1, we can derive the fixed point condition for DAG-Adam as follows (see the derivation in Appendix C):

$$\nabla f_i(x_i^\infty) + \frac{\nu}{\gamma}(x_i^\infty - \sum_{j \in \mathcal{N}_i} w_{ij} x_j^\infty) = 0, \ \ \forall i \in [n]. \tag{15}$$

$$\implies \ \frac{1}{n}\sum_{i=1}^n \nabla f_i(x^\infty) = 0 \qquad (\text{if } x_1^\infty = \cdots = x_n^\infty = x^\infty \text{ in the limit}) \tag{16}$$

Note that the fixed-point condition (15) for DAG-Adam, when $\nu = 1$, is exactly the same as that for vanilla decentralized SGD with full-bath gradient (Yuan et al., 2016). Relation (16) implies that DAG-Adam will converge to the stationary solution to problem (1). The following proposition characterizes the distance between the DAG-Adam fixed point and the global optimal solution $x^\star$ for the strongly-convex scenario.

**Proposition 3** *Under the same conditions as in Proposition 1, the distance between DAG-Adam fixed point and $x^\star$ can be characterized as follows:*

$$\lim_{k \to \infty} \sum_{i=1}^n \|x_i^{(t)} - x^\star\|^2 = O\Big(\frac{\gamma^2 G^2}{(1-\rho)^2 \nu^2}\Big) \tag{17}$$

It is observed in (17) that the distance is on the order of $O(\gamma^2)$. Apparently, DAG-Adam will converge to the global optimal solution when $\gamma \to 0$. Moreover, DAG-Adam has the same communication overhead as in DAdam, and it is more communication-efficient than GT-DAdam. Proposition 3 also implies that DAG-Adam has the same limiting bias as decentralized SGD when $\nu = 1$.

**Limiting point justification.** We validate the limiting point analysis for DAdam, QG-DAdam, GT-DAdam, and DAG-Adam on the full-batch logistic regression problem. Table 5 in Appendix C.3 lists the results we establish in Propositions 1 – 4. When $\nu = 1$ in the DAG-DAdam method, which is a valid choice in convex settings in numerical experiments, the fixed point of DAG-DAdam is closet to the global solution $x^\star$ when the cost function is strongly-convex. The learning curves of

DAdam, GT-DAdam, QG-DAdam, and DAG-Adam are shown in Fig. 1. The MSE metric in the $y$-axis denotes $\frac{1}{n}\sum_{i=1}^{n}\|x_i^{(t)} - x^\star\|^2$. From the figure, it is observed that the DAG-Adam algorithm can converge as fast as other decentralized adaptive algorithms but to a more accurate limiting point. This limiting point is exactly the same as vanilla decentralized SGD, which is consistent with our claim in the limiting point analysis. More experimental setting and results refer to Appendix C.4.

**Convergence Analysis.** We provide the convergence analysis for our proposed DAG-Adam for non-convex functions. The full proof can be found in Appendix E.

**Theorem 1** *Suppose the parameters $\gamma$, $\nu$, and $\beta_2$ satisfy the conditions that $\gamma \leq \frac{\epsilon - 2\nu}{2L}$, $\nu < \min\{\epsilon/2, \kappa\gamma\}$, ($\kappa$ is any positive constant) and $1 - \beta_2 \leq \frac{\epsilon^2}{16(G+\kappa C)^2}$. Under Assumptions A.1 - A.4 and the augmented gradient shown is bounded, the decentralized augmented gradient generated by DAG-Adam algorithm satisfies that*

$$\frac{1}{T}\sum_{t=0}^{T-1}\sum_{i=1}^{n}\left\|\nabla f_i(x_i^{(t)}) + \frac{\nu}{\gamma}(x_i^{(t)} - \sum_{j\in\mathcal{N}_i} w_{ij}x_j^{(t)})\right\|^2 = O\left(\frac{\mathcal{L}(\{x_i^{(0)}\}_{i=0}^{n}) - \mathcal{L}(\mathbf{x}^*)}{\gamma T}\right) + O(\sigma^2) \tag{18}$$

*where the augmented loss function $\mathcal{L}(\mathbf{x})$ is the one defined in (14), $\mathcal{L}^\star$ is the minimum of the augmented loss. Further, it holds for the globally averaged iterate $\bar{x}^{(t)}$ that*

$$\frac{1}{T}\sum_{t=0}^{T-1}\|\nabla f(\bar{x}^{(t)})\|^2 \leq O\left(\frac{\mathcal{L}(\{x_i^{(0)}\}_{i=0}^{n}) - \mathcal{L}^\star}{\gamma T}\right) + O(\sigma^2) + O\left(\frac{\gamma^2}{(1-\rho)^2\nu^2}\right) \tag{19}$$

*where $\bar{x}^{(t)} = 1/n\sum_{i=1}^{n}x_i^{(t)}$ and $\rho$ is the second largest eigenvalue of $W$ in magnitude.*

**Remark 1** *The theorem stated two conclusions. The first one is that the augmented gradient in DAG-Adam will converge to a neighborhood around 0 with constant learning rate $\gamma$, and the neighborhood, when $T \to \infty$, is on the order of $O(\sigma^2)$. The second one is that DAG-Adam will converge to a neighborhood around the stationary solution in terms of the globally averaged iterate $\bar{x}^{(t)}$. The neighborhood, when learning rate $\gamma = O(1/\sqrt{T})$ and $T \to \infty$, will be on the order of $O(\sigma^2)$. It is thus observed that DAG-Adam will converge to the exact stationary solution with decaying learning rate and increasing mini-batch sizes, which is consistent with the results of Adam using the standard stochastic gradient in (Zaheer et al., 2018).*

**Remark 2** *The non-vanishing term $O(\sigma^2)$ in (18) and (19) is caused by the vanilla Adam algorithm structure, not the decentralized communication and update in DAG-Adam. Such term can be corrected by using AMSGrad rather than Adam in DAG-Adam. When there is no stochastic gradient noise, the augmented gradient will converge to zero. In consequence, the $v_i^{(t)}$ will converge to zero over all agents as well.*

## 4 EXPERIMENTS

In this section, we systematically compare the proposed method, DAG-Adam, with all previous mentioned state-of-the-art decentralized adaptive methods on typical CV and NLP tasks.

**Implementation Details.** We implement all the aforementioned algorithms with PyTorch 1.8.0 (Paszke et al. (2019)) using NCCL 2.9.6 (CUDA 10.2) as the communication backend. For PAdam, we used PyTorch's native Distributed Data Parallel (DDP) module. For the implementation of decentralized methods, we utilize BlueFog. See Appendix F for more details.

### 4.1 IMAGE CLASSIFICATION: PERFORMANCE OVER VARIOUS CONFIGURATIONS

A series of experiments are carried out with CIFAR-10 (Krizhevsky et al. (2009)) and ImageNet (Deng et al. (2009)) datasets to evaluate the performance of the proposed DAG-Adam on both homogeneous and heterogeneous dataset over different sizes of network. We utilize ResNet-20 on CIFAR-10 and ResNet-50 on ImageNet (He et al., 2016).

**Training configuration.** All methods are trained using 90 epochs with mini-batch of size 32 in each computing node. PAdam uses a learning rate of 0.0125, while the ones for decentralized algorithms are scaled by the network size. In addition, QG-Adam (Lin et al. (2021)) uses a learning rate 10 times smaller than other decentralized methods to avoid divergence[1]. During the training procedure,

---

[1]We examine QG-DAdam based on our own implementation of the algorithm listed in the original paper.

Table 1: Classification accuracy comparison of different methods for CIFAR10 dataset.

| Methods | 4 Nodes | 8 Nodes | | | |
|---|---|---|---|---|---|
| | IID | $\alpha = 0.1$ | $\alpha = 1$ | $\alpha = 10$ | IID |
| PADAM | 90.28 ±0.07 | 51.65 ±0.88 | 84.33 ±0.18 | 90.82 ±0.23 | 91.19 ±0.09 |
| DADAM | 87.23 ±0.14 | 53.71 ±1.38 | 83.67 ±0.14 | 86.34 ±0.51 | 87.04 ±0.12 |
| QG-DADAM | 83.43 ±0.22 | 34.24 ±2.19 | 57.89 ±0.34 | 71.97 ±0.49 | 77.82 ±0.13 |
| GT-DADAM | 86.93 ±0.17 | **56.48 ±1.26** | 84.63 ±0.22 | 86.71 ±0.31 | 86.47 ±0.12 |
| DAG-ADAM ($\nu = 10^{-3}$) | 84.62 ±0.49 | 46.77 ±1.09 | 75.76 ±0.34 | 81.88 ±0.79 | 82.38 ±0.55 |
| DAG-ADAM ($\nu = 10^{-2}$) | 87.50 ±0.36 | 51.63 ±1.07 | 82.24 ±1.06 | 87.82 ±0.35 | 88.55 ±0.33 |
| DAG-ADAM ($\nu = 10^{-1}$) | **91.39 ±0.09** | 55.27 ±0.31 | **86.64 ±0.03** | **91.00 ±0.18** | **91.23 ±0.09** |

the learning rate is warmed up in the first 5 epochs and is then decayed by a factor of 10 at 30, 60 and 80 epochs. The weight decay is chosen as $5 \times 10^{-5}$. We also compared the results of the proposed DAG-Adam with various $\nu$ choices.

**Heterogenerous data configuration.** To simulate the heterogenerous data, the disjoint non-i.i.d. training data over agents are generated through Dirichlet distribution $\mathrm{Dir}(\alpha)$, where $\alpha$ stands for the degree of the data heterogeneity (Lin et al., 2021; Yurochkin et al., 2019). The training data for a particular class tends to concentrate in a single node as $\alpha \to 0$, i.e. becoming more heterogenous while the homogeneous data is formed as $\alpha \to \infty$. See Appendix F for more details.

**Performance comparison.** Table 1 compares the proposed DAG-Adam with other methods over networks with both homogeneous data distribution and heterogeneous ones. Two different network sizes (4 and 8) are tested using ring topology. With a proper choice of $\nu$, the proposed method *consistently* outperforms all other methods under comparison, including the centralized optimization method PAdam, no matter what the data distribution is given. The left and middle figures in Fig. 2 shows how the test loss and test accuracy evolves in a heterogeneous data distribution with $\alpha = 10$. DAG-Adam (with $\nu = 1e-1$) converges faster than any other decentralized algorithms. As typical decentralized algorithm behaves, it was relatively slower compared with centralized PAdam algorithm in the beginning, but it eventually caught up when the algorithm converged. The right figure in Fig. 2 plots the consensus error $\sum_i \|x_i - \bar{x}\|$, which validates that the consensus error in Theorem 1 is proportional to the choice of $\nu$.

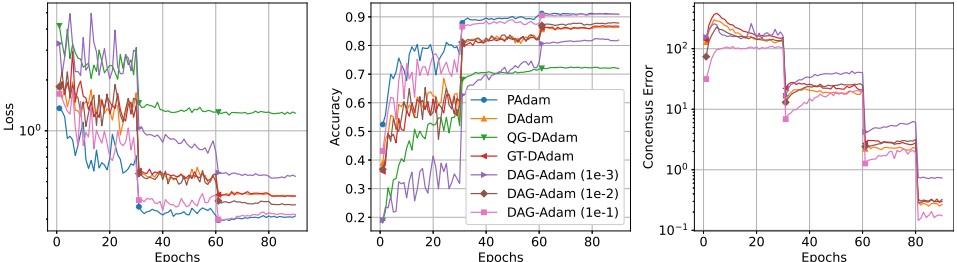

Figure 2: The evolution comparison of loss, accuracy and consensus error for CIFAR-10 dataset on a heterogeneous ($\alpha = 10$) network of size 8.

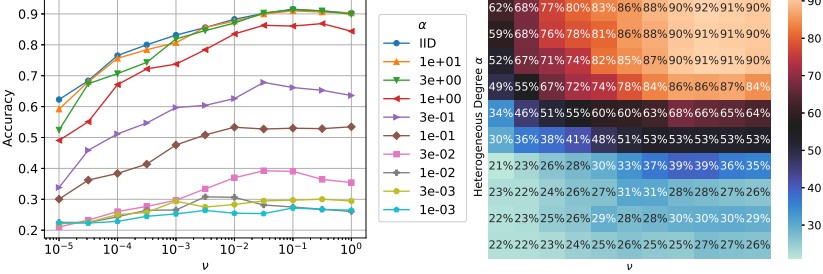

Figure 3: An ablation study of the hyperparameter $\nu$ in DAG-Adam for CIFAR-10 dataset on heterogeneous networks of size 8. The heatmap on the right shares the same selection of $\nu$ and $\alpha$ with the left figure. We reported the detailed accuracy value in it.

**Sensitivity on $\nu$ constant.** As shown in Table 1, the final classification performance depends on the choice of constant $\nu$ in DAG-Adam. To better understand how $\nu$ affects the DAG-Adam performance, we performed a corresponding ablation study with varying heterogeneity degree $\alpha$ as shown

in Fig. 3. For varying heterogeneity degree $\alpha$, the constant $\nu$ that achieving best performance lies in the range of $[1e^{-2}, 5e^{-1}]$ for CIFAR-10 typically.

**Performance over different network scales.** We further examine the performance of the proposed algorithm over different network sizes to see how good it can scale up in large decentralized network case. As shown in Table 2, under the exponential graph (Assran et al., 2019) and fine-tuning of $\nu$, DAG-Adam persistently achieves comparable performance with the centralized method until the node size reaches $n = 64$. Even under that case, the drop of performance is quite small.

Table 2: Comparison of different nodes and methods on CIFAR-10 classification over homogeneous data.

| NODES | 4 | 8 | 16 | 32 | 64 |
|---|---|---|---|---|---|
| PADAM | 91.28 ±0.07 | 91.19 ±0.09 | 90.67 ±0.09 | 90.15 ±0.11 | **90.11 ±0.13** |
| DAG-ADAM ($\nu = 1e-3$) | 84.62 ±0.57 | 82.38 ±0.55 | 84.35 ±0.51 | 83.73 ±0.49 | 83.90 ±0.61 |
| DAG-ADAM ($\nu = 1e-2$) | 87.50 ±0.29 | 86.55 ±0.33 | 86.89 ±0.42 | 87.83 ±0.31 | 87.62 ±0.37 |
| DAG-ADAM ($\nu = 1e-1$) | **91.39 ±0.11** | **91.23 ±0.09** | **91.36 ±0.12** | **90.16 ±0.07** | 89.98 ±0.22 |

**ImageNet results.** We also validate the performance of our proposed method on a relatively large-scale dataset ImageNet. We use PAdam to tune the learning rate ($4 \times 10^{-4}$) and wight decay ($1 \times 10^{-4}$) hyper-parameters to reach a strong baseline. Then, the same hyper-parameters are applied for decentralized counterparts. In the experiment, the algorithms run on 8 nodes (i.e. $8 \times 8 = 64$ GPUs) and train total 100 epochs with a cosine decay scheduler. Our proposed method reaches top-1 accuracy $75.96\%$ which is slightly better than PAdam.

Table 3: Comparison of different methods and models on ImageNet classification.

| METHOD | PADAM | DADAM | QG | GT | DAG/$1e-1$ | DAG/$1e-2$ | DAG/$1e-3$ | DAG/$1e-4$ |
|---|---|---|---|---|---|---|---|---|
| ACCURACY | 75.62 | 70.53 | 73.89 | 74.736 | 66.506 | 73.750 | 75.445 | **75.96** |

## 4.2 Language Modeling: Performance over different tasks

BERT (Devlin et al. (2018)) is a widely used pre-training language representation model for NLP tasks. We fine-tuned pretrained BERT models (Base/Large: 110/330M parameters) on the SQuAD (Rajpurkar et al. (2016)) dataset. All experiments are based on NVIDIA BERT[2] implementation with mixed precision support. We examine the task over both static exponential graph and dynamic exponential graph setting (Assran et al., 2019). As shown in Table 4, with proper tuned hyper-parameter $\nu$, our proposed method reached best performance in both metrics.

Table 4: Comparison of different methods on BERT fine-tuning (SQuAD) on Exact Match and F1-score.

| MODEL | BERT-BASE | | | | BERT-LARGE | | | |
|---|---|---|---|---|---|---|---|---|
| TOPOLOGY | STATIC | | DYNAMIC | | STATIC | | DYNAMIC | |
| EM | F1 | EM | F1 | EM | F1 | EM | F1 | EM |
| PADAM | 81.50 | 88.85 | - | - | 83.54 | 90.69 | - | - |
| DADAM | 80.61 | 87.57 | 80.31 | 87.42 | 83.25 | 89.98 | 83.16 | 89.77 |
| QG-DADAM | 79.4 | 86.98 | 80.17 | 87.33 | 83.05 | 89.84 | 83.89 | 90.64 |
| GT-DADAM | 80.66 | 87.85 | 80.67 | 87.81 | 83.81 | 90.51 | 83.95 | 90.61 |
| DAG-ADAM ($\nu = 1e-1$) | 48.89 | 60.85 | 43.77 | 56.22 | 59.24 | 71.40 | 55.18 | 67.89 |
| DAG-ADAM ($\nu = 1e-2$) | 71.98 | 81.62 | 70.30 | 80.15 | 78.20 | 86.68 | 76.85 | 85.59 |
| DAG-ADAM ($\nu = 1e-3$) | 80.66 | 87.99 | 80.14 | 87.54 | **83.93** | 90.48 | 83.31 | 90.26 |
| DAG-ADAM ($\nu = 1e-4$) | **81.79** | **88.93** | **81.89** | **88.96** | 83.82 | **90.71** | **84.28** | **91.08** |

## 5 Conclusion and future works

In this paper, we identified the limitation of adapt-while/then-communicate structure utilized in existing adaptive gradient methods: the limiting discrepancy in the squared local gradient term (i.e., $v_i^\infty$) can deviate the limiting point from the desired stationary point. To fix this issue, we proposed a novel DAG-Adam algorithm that is built upon the communicate-then-adapt structure instead. DAG-Adam can provably converge to the desired solution even with heterogeneous data. Experimental results on a variety of deep learning tasks show that DAG-Adam persistently outperforms existing decentralized adaptive baselines. However, it is also observed in these experiments that DAG-Adam is highly-sensitive to the hyper-parameter $\nu$. How to tune $\nu$ automatically during the training stage is a crucial topic of the future works.

---

[2]https://github.com/NVIDIA/DeepLearningExamples/tree/master/PyTorch/LanguageModeling/BERT

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

## A  NOTATIONS AND PRELIMINARIES

**Notations.** We introduce several notations as follows:

- $\mathbf{x} \triangleq \begin{bmatrix} \boldsymbol{x}_1 \\ \boldsymbol{x}_2 \\ \vdots \\ \boldsymbol{x}_n \end{bmatrix} \in \mathbb{R}^{nd \times 1}, \quad \mathbf{m} \triangleq \begin{bmatrix} \boldsymbol{m}_1 \\ \boldsymbol{m}_2 \\ \vdots \\ \boldsymbol{m}_n \end{bmatrix} \in \mathbb{R}^{nd \times 1}, \quad \mathbf{v} \triangleq \begin{bmatrix} \boldsymbol{v}_1 \\ \boldsymbol{v}_2 \\ \vdots \\ \boldsymbol{v}_n \end{bmatrix} \in \mathbb{R}^{nd \times 1}.$

- $\nabla \widehat{\mathbf{f}}(\mathbf{x}) \triangleq \begin{bmatrix} \nabla F_1(\boldsymbol{x}_1; \xi_1) \\ \nabla F_2(\boldsymbol{x}_2; \xi_2) \\ \vdots \\ \nabla F_n(\boldsymbol{x}_n; \xi_n) \end{bmatrix} \in \mathbb{R}^{nd \times 1}.$

- $W = [w_{ij}] \in \mathbb{R}^{n \times n}.$

- $\mathcal{W} \triangleq W \otimes I_d \in \mathbb{R}^{nd \times nd}$, where $\otimes$ means the Kronecker product.

- $\mathbb{1}_n = \mathrm{col}\{1, 1, \ldots, 1\} \in \mathbb{R}^n.$

- $\bar{\mathbf{x}} = (\frac{1}{n}\mathbb{1}_n\mathbb{1}_n^T \otimes I_d)\mathbf{x} \in \mathbb{R}^{nd \times 1}.$

- Given a vector $x \in \mathbb{R}^d$, we let $\mathrm{diag}(x) = \mathrm{diag}\{x_1, \cdots, x_d\} \in \mathbb{R}^{d \times d}$ be a diagonal matrix.

- Given a positive definite matrix $A$, we define the weighted normal $\|x\|_A^2 = x^T A x$, $\|x\|$ as vector $\ell_2$ norm and $\|A\|$ as matrix $\ell_2$ norm.

**Algorithm reformulation.** With the above notation, we rewrite DAG-Adam in Algorithm 1 into the following stacked vector form:

$$\mathbf{g}^{(t)} = \nabla \widehat{\mathbf{f}}(\mathbf{x}^{(t)}) + \frac{\nu}{\gamma}(I - \mathcal{W})\mathbf{x}^{(t)} \tag{20}$$

$$\mathbf{m}^{(t)} = \beta_1 \mathbf{m}^{(t-1)} + (1 - \beta_1)\mathbf{g}^{(t)} \tag{21}$$

$$\mathbf{v}^{(t)} = \beta_2 \mathbf{v}^{(t-1)} + (1 - \beta_2)\mathbf{g}^{(t)} \odot \mathbf{g}^{(t)} \tag{22}$$

$$\mathbf{x}^{(t+1)} = \mathbf{x}^{(t)} - \gamma(\mathbf{H}^{(t)} + \epsilon I)^{-1}\mathbf{m}^{(t)} \tag{23}$$

where $\mathbf{H}^{(t)}$ is the diagonal matrix defined as $\mathbf{H}^{(t)} = \mathrm{Diag}(\sqrt{\mathbf{v}^{(t)}})$.

**Smoothness.** Since each $f_i(x)$ is assumed to be $L$-smooth in Assumption A.1, it holds that $f(x) = \frac{1}{n}\sum_{i=1}^n f_i(x)$ is also $L$-smooth. As a result, the following inequality holds for any $\boldsymbol{x}, \boldsymbol{y} \in \mathbb{R}^d$:

$$f(\boldsymbol{x}) - f(\boldsymbol{y}) - \frac{L}{2}\|\boldsymbol{x} - \boldsymbol{y}\|^2 \leq \langle \nabla f(\boldsymbol{y}), \boldsymbol{x} - \boldsymbol{y} \rangle \tag{24}$$

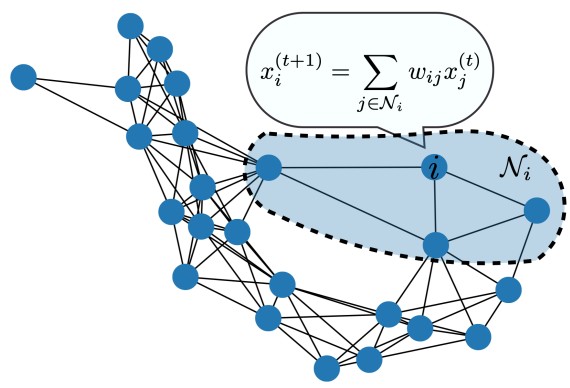

Figure 4: An illustration of decentralized topology and its communication pattern

---

**Algorithm 2** DAdam (Nazari et al., 2019)

---

**Initialize** $x_i^{(0)}$ arbitrarily; let $\beta_1, \beta_2, \beta_3 \in (0,1)$; let $m_i^{(0)} = v_i^{(0)} = \hat{v}_i^{(0)} = 0$; set $\gamma$ properly;
**For** $t = 0, 1, 2, ..., T-1$, every node $i$ **do**

    Sample $g_i^{(t)} = \nabla F(x_i^{(t)}; \xi_i^{(t)})$;

    $m_i^{(t)} = \beta_1 m_i^{(t-1)} + (1-\beta_1) g_i^{(t)}$;

    $v_i^{(t)} = \beta_2 v_i^{(t-1)} + (1-\beta_2) g_i^{(t)} \odot g_i^{(t)}$;

    $\hat{v}_i^{(t)} = \beta_3 \hat{v}_i^{(t-1)} + (1-\beta_3) \max\{\hat{v}_i^{(t-1)}, v_i^{(t)}\}$;

    $x_i^{(t+1)} = \sum_{j \in \mathcal{N}_i} w_{ij} x_j^{(t)} - \gamma \frac{m_i^{(t)}}{\sqrt{\hat{v}_i^{(t)} + \epsilon}}$;

---

**Weighted norm inequality.** In this paper, the weighted norm $\|x\|_A^2$ only considers the scenario in which $A$ is a diagonal matrix. Suppose the diagonal elements in the diagonal matrix $A$ satisfy that $0 < a \le [A]_{i,i} \le b$ for all $i$, one useful inequality is

$$a\|x\|_A^2 \le \|Ax\|^2 \le b\|x\|_A^2 \tag{25}$$

The proof is strait-forward:

$$\|Ax\|^2 = x^T A^2 x = (A^{1/2}x)^T A(A^{1/2}x) \le b(A^{1/2}x)^T(A^{1/2}x) = b\|x\|_A^2 \tag{26}$$

We can use the same argument to show $a\|x\|_A^T \le \|Ax\|^2$ as well.

**Communication topology.** Figure 4 provides a graph example to illustrate the decentralized topology and its communication pattern that is used in all decentralized methods listed in the paper.

## B ADAPT-WHILE/THEN-COMMUNICATE STRUCTURE AND ITS LIMITATION

### B.1 DADAM AND ADAPT-WHILE-COMMUNICATE STRUCTURE

The DAdam algorithm (Nazari et al., 2019) is listed in Algorithm 2.

### B.2 FIXED POINT CONDITION FOR DADAM

To see the fixed point condition for DAdam, we assume the iterate $x_i^{(t)}$ in DAdam converges to some fixed point $x_i^\infty$ as $k \to \infty$ for any $i \in [n]$. In order to remove the influence of the gradient noise, we let $g_i^{(t)}$ be the full-batch gradient $\nabla f_i(x_i^{(t)}) = \mathbb{E}[\nabla F(x_i^{(t)}; \xi_i)]$ for any $i \in [n]$. Under this setting, it is easy to verify that

$$m_i^\infty = \nabla f_i(x_i^\infty), \quad \text{and} \quad v_i^\infty = \hat{v}_i^\infty = \nabla f_i(x_i^\infty) \odot \nabla f_i(x_i^\infty) \tag{27}$$

If we define $H_i := \text{diag}(\sqrt{\nabla f_i(x_i^\infty) \odot \nabla f_i(x_i^\infty)})$ to be a positive diagonal matrix, then the highlighted step in DAdam will become (when $t \to \infty$)

$$x_i^\infty = \sum_{j \in \mathcal{N}_i} w_{ij} x_j^\infty - \gamma(H_i + \epsilon I)^{-1} \nabla f_i(x_i^\infty), \quad \forall i \in [n]. \tag{28}$$

which is the fixed-point condition listed in (5). If we further let $\bar{x}^\infty := \frac{1}{n}\sum_{i=1}^n x_i^\infty$, the following relation can be achieved by (28) and Assumption A.3:

$$\bar{x}^\infty = \bar{x}^\infty - \frac{\gamma}{n}\sum_{i=1}^n (H_i + \epsilon I)^{-1} \nabla f_i(x_i^\infty)$$

$$\implies \frac{1}{n}\sum_{i=1}^n (H_i + \epsilon I)^{-1} \nabla f_i(x_i^\infty) = 0$$

$$\implies \frac{1}{n}\sum_{i=1}^n (H_i + \epsilon I)^{-1} \nabla f_i(x^\infty) = 0 \quad (\text{if } x_1^\infty = \cdots = x_n^\infty = x^\infty \text{ in the limit}). \tag{29}$$

The last relation is the one listed in (6).

### B.3 Proof of Proposition 1

Before we prove Proposition 1, we need to introduce several notations:

- $\mathbf{x}^\infty = \text{col}\{x_1^\infty, \cdots, x_n^\infty\} \in \mathbb{R}^{nd}$
- $\bar{\mathbf{x}}^\infty = \text{col}\{\bar{x}^\infty, \cdots, \bar{x}^\infty\} \in \mathbb{R}^{nd}$ where $\bar{x}^\infty = \frac{1}{n}\sum_{i=1}^n x_i^\infty$.
- $\mathbf{x}^\star = \text{col}\{x^\star, \cdots, x^\star\} \in \mathbb{R}^{nd}$ where $x^\star$ is the optimal solution to problem (1) when the cost function is convex
- $\nabla \mathbf{f}(\mathbf{x}^\infty) = \text{col}\{\nabla f_1(x_1^\infty), \cdots, \nabla f_n(x_n^\infty)\} \in \mathbb{R}^{nd}$
- $(\mathbf{H} + \epsilon I)^{-1} = \text{diag}\{(H_1 + \epsilon I)^{-1}, \cdots, (H_n + \epsilon I)^{-1}\}$

With these notations, we can rewrite the fixed-point condition (28) into a compact manner:

$$\mathbf{x}^\infty = \mathcal{W}\mathbf{x}^\infty - \gamma(\mathbf{H} + \epsilon I)^{-1}\nabla \mathbf{f}(\mathbf{x}^\infty) \tag{30}$$

To examine $\|\mathbf{x}^\infty - \mathbf{x}^\star\|$, we will evaluate $\|\mathbf{x}^\infty - \bar{\mathbf{x}}^\infty\|$ and $\|\bar{x}^\infty - x^\star\|$ respectively.

**Upper bound of** $\|\bar{x}^\infty - x^\star\|$. Left-multiplying $\frac{1}{n}\mathbb{1}^T$ to both sides of recursion (30), we achieve

$$\frac{1}{n}\sum_{i=1}^n (H_i + \epsilon I)^{-1}\nabla f_i(x_i^\infty) = 0. \tag{31}$$

With the above fact, we have

$$\|\bar{x}^\infty - x^\star\|$$
$$= \|\bar{x}^\infty - x^\star - \gamma\left(\frac{1}{n}\sum_{i=1}^n (H_i + \epsilon I)^{-1}\nabla f_i(x_i^\infty) - \frac{1}{n}\sum_{i=1}^n \nabla f_i(x^\star)\right)\|$$
$$\leq \|\bar{x}^\infty - x^\star - \gamma\left(\frac{1}{n}\sum_{i=1}^n \nabla f_i(\bar{x}^\infty) - \frac{1}{n}\sum_{i=1}^n \nabla f_i(x^\star)\right)\|$$
$$+ \gamma\|\frac{1}{n}\sum_{i=1}^n (H_i + \epsilon I)^{-1}\nabla f_i(x_i^\infty) - \frac{1}{n}\sum_{i=1}^n \nabla f_i(x_i^\infty)\|$$
$$+ \gamma\|\frac{1}{n}\sum_{i=1}^n \nabla f_i(x_i^\infty) - \frac{1}{n}\sum_{i=1}^n \nabla f_i(\bar{x}^\infty)\|$$
$$\leq (1 - \frac{\gamma\mu}{2})\|\bar{x}^\infty - x^\star\| + \gamma B + \frac{\gamma L}{\sqrt{n}}\|\mathbf{x}^\infty - \bar{\mathbf{x}}^\infty\| \tag{32}$$

where the last inequality holds because each $f_i(x)$ is $\mu$-strongly convex and $L$-smooth, and the definition of constant $B$ as follows:

$$B := \|\frac{1}{n}\sum_{i=1}^n (H_i + \epsilon I)^{-1}\nabla f_i(x_i^\infty) - \frac{1}{n}\sum_{i=1}^n \nabla f_i(x_i^\infty)\|. \tag{33}$$

Inequality (32) leads to

$$\sqrt{n}\|\bar{x}^\infty - x^\star\| \leq \frac{2\sqrt{n}B}{\mu} + \frac{2L}{\mu}\|\mathbf{x}^\infty - \bar{\mathbf{x}}^\infty\|. \tag{34}$$

**Upper bound of** $\|\mathbf{x}^\infty - \bar{\mathbf{x}}^\infty\|$. Subtracting $\bar{\mathbf{x}}^\infty$ from both sides of (30), we have

$$(I - \mathcal{W})(\mathbf{x}^\infty - \bar{\mathbf{x}}^\infty) = -\gamma(\mathbf{H} + \epsilon I)^{-1}\nabla \mathbf{f}(\mathbf{x}^\infty). \tag{35}$$

With inequality (43) in (Yuan et al., 2021), we have

$$\|(I - \mathcal{W})(\mathbf{x}^\infty - \bar{\mathbf{x}}^\infty)\| \geq (1 - \rho)\|\mathbf{x}^\infty - \bar{\mathbf{x}}^\infty\| \tag{36}$$

With (35) and (36), and noticing $\|(\mathbf{H} + \epsilon I)^{-1}\| = \max_i\{\|(H_i + \epsilon I)^{-1}\|_2\} \leq \frac{1}{\epsilon}$, we achieve

$$\|\mathbf{x}^\infty - \bar{\mathbf{x}}^\infty\| \leq \frac{\gamma\|\nabla \mathbf{f}(\mathbf{x}^\infty)\|}{(1 - \rho)\epsilon} \leq \frac{\gamma G}{(1 - \rho)\epsilon} \tag{37}$$

where the last inequality holds because of Assumption A.4.

**Upper bound of** $\lim_{t\to\infty} \frac{1}{n}\sum_{i=1}^{n} \|x_i^{(t)} - x^\star\|^2$. With (34) and (37), we have

$$
\begin{aligned}
\|\mathbf{x}^\infty - \mathbf{x}^\star\| &\le \|\mathbf{x}^\infty - \bar{\mathbf{x}}^\infty\| + \sqrt{n}\|\bar{x}^\infty - x^\star\| \\
&\overset{(34)}{\le} (1 + \frac{2L}{\mu})\|\mathbf{x}^\infty - \bar{\mathbf{x}}^\infty\| + \frac{2\sqrt{n}B}{\mu} \\
&\overset{(37)}{\le} (1 + \frac{2L}{\mu})\frac{\gamma G}{(1-\rho)\epsilon} + \frac{2\sqrt{n}B}{\mu}.
\end{aligned}
\tag{38}
$$

If we ignore the influence of $\mu$ and $L$, we finally achieve

$$
\lim_{t\to\infty} \frac{1}{n}\sum_{i=1}^{n} \|x_i^{(t)} - x^\star\|^2 = \frac{1}{n}\|\mathbf{x}^\infty - \mathbf{x}^\star\|^2 = O\Big(\frac{\gamma^2 G^2}{(1-\rho)^2\epsilon^2} + B^2\Big).
\tag{39}
$$

**Lower bound of** $\lim_{t\to\infty} \frac{1}{n}\sum_{i=1}^{n} \|x_i^{(t)} - x^\star\|^2$. Now we show that the terms depending on $B^2$ in the upper bound (39) are necessary and cannot be removed by an improved analysis. In fact, there exists strongly convex and smooth functions such that $x_i^{(t)}$ generated by DAdam satisfies

$$
\lim_{t\to\infty} \frac{1}{n}\sum_{i=1}^{n} \|x_i^{(t)} - x^\star\|^2 \ge B^2.
\tag{40}
$$

To prove the above conclusion, we consider minimizing problem of the form (1) with

$$
f_i(x) = \frac{1}{2}(x - y_i)^2, \quad \forall i \in [n]
\tag{41}
$$

where $x, y_i \in \mathbb{R}$. Apparently, the solution to problem (1) is given by $x^\star = \frac{1}{n}\sum_{i=1}^{n} y_i$. Next we let $x_i^\infty$ be the limiting point of DAdam (Algorithm 2) in node $i$. With notations in Sec. B.2, we have

$$
\nabla f_i(x_i^\infty) = x_i^\infty - y_i
\tag{42}
$$
$$
H_i = |\nabla f_i(x_i^\infty)| = |x_i^\infty - y_i|
\tag{43}
$$
$$
(\mathbf{H} + \epsilon)^{-1} = \text{diag}\{(H_i + \epsilon)^{-1}\} \in \mathbb{R}^{n\times n}
\tag{44}
$$

Under the above setting, the limiting point of DAdam (30) reduces to

$$
\mathbf{x}^\infty = W\mathbf{x}^\infty - \gamma(\mathbf{H} + \epsilon I)^{-1}\nabla f(\mathbf{x}^\infty)
\tag{45}
$$

in which $W \in \mathbb{R}^{n\times n}$ is the weight matrix. By left-multiplying $\frac{1}{n}\mathbb{1}^T$ to both sides, we have

$$
\frac{1}{n}\sum_{i=1}^{n} \frac{\nabla f_i(x_i^\infty)}{|\nabla f_i(x_i^\infty)| + \epsilon} = \frac{1}{n}\sum_{i=1}^{n} \frac{x_i^\infty - y_i}{|x_i^\infty - y_i| + \epsilon} = 0.
\tag{46}
$$

On the other hand, it is easy to verify that

$$
\frac{1}{n}\sum_{i=1}^{n} \nabla f_i(\bar{x}^\infty) = \frac{1}{n}\sum_{i=1}^{n} (\bar{x}^\infty - y_i) = \bar{x}^\infty - \frac{1}{n}\sum_{i=1}^{n} y_i = \bar{x}^\infty - x^\star.
\tag{47}
$$

With (46) and (47), we have

$$
\begin{aligned}
\|\bar{x}^\infty - x^\star\| &\overset{(47)}{=} \|\frac{1}{n}\sum_{i=1}^{n} \nabla f_i(\bar{x}^\infty)\| \\
&\overset{(46)}{=} \|\frac{1}{n}\sum_{i=1}^{n} \frac{\nabla f_i(x_i^\infty)}{|\nabla f_i(x_i^\infty)| + \epsilon} - \frac{1}{n}\sum_{i=1}^{n} \nabla f_i(\bar{x}^\infty)\| \\
&= \|\frac{1}{n}\sum_{i=1}^{n} \frac{\nabla f_i(x_i^\infty)}{|\nabla f_i(x_i^\infty)| + \epsilon} - \frac{1}{n}\sum_{i=1}^{n} \nabla f_i(x_i^\infty) + \frac{1}{n}\sum_{i=1}^{n} \nabla f_i(x_i^\infty) - \frac{1}{n}\sum_{i=1}^{n} \nabla f_i(\bar{x}^\infty)\|
\end{aligned}
$$

---

**Algorithm 3** QG-DAdam (Lin et al., 2021)

---

**Initialize** $x_i^{(0)}$ arbitrarily; let $\beta_1, \beta_2 \in (0,1)$; let $m_i^{(0)} = \hat{m}_i^{(0)} = v_i^{(0)} = \hat{v}_i^{(0)} = 0$; set $\gamma$ properly;
**For** $k = 0, 1, 2, ..., T-1$, every node $i$ **do**

  Sample $g_i^{(t)} = \nabla F(x_i^{(tk)}; \xi_i^{(t)})$;
  $m_i^{(t)} = \beta_1 \hat{m}_i^{(t-1)} + (1-\beta_1)g_i^{(t)}$;
  $v_i^{(t)} = \beta_2 \hat{v}_i^{(t-1)} + (1-\beta_2)g_i^{(t)} \odot g_i^{(t)}$;
  $x_i^{(t+1)} = \sum_{j \in \mathcal{N}_i} w_{ij}\big(x_j^{(t)} - \gamma \frac{m_j^{(t)}}{\sqrt{v_j^{(t)}} + \epsilon}\big)$;
  $d_i^{(t)} = (x_i^{(t+1)} - x_i^{(t)})/\|x_i^{(t+1)} - x_i^{(t)}\|_2$;
  $\hat{m}_i^{(t)} = \beta_1 \hat{m}_i^{(t-1)} + (1-\beta_1)d_i^{(t)}$;
  $\hat{v}_i^{(t)} = \beta_2 \hat{v}_i^{(t-1)} + (1-\beta_2)d_i^{(t)} \odot d_i^{(t)}$;

---

$$\overset{(42)}{=} \|\frac{1}{n}\sum_{i=1}^n \frac{\nabla f_i(x_i^\infty)}{|\nabla f_i(x_i^\infty)| + \epsilon} - \frac{1}{n}\sum_{i=1}^n \nabla f_i(x_i^\infty) + \frac{1}{n}\sum_{i=1}^n (x_i^\infty - \bar{x}^\infty)\|$$

$$= \|\frac{1}{n}\sum_{i=1}^n \frac{\nabla f_i(x_i^\infty)}{|\nabla f_i(x_i^\infty)| + \epsilon} - \frac{1}{n}\sum_{i=1}^n \nabla f_i(x_i^\infty)\|$$

$$\overset{(43)}{=} \|\frac{1}{n}\sum_{i=1}^n (H_i + \epsilon I)^{-1}\nabla f_i(x_i^\infty) - \frac{1}{n}\sum_{i=1}^n \nabla f_i(x_i^\infty)\| = B. \tag{48}$$

Finally, we have

$$\lim_{t\to\infty} \frac{1}{n}\sum_{i=1}^n \|x_i^{(t)} - x^\star\|^2 = \frac{1}{n}\|\mathbf{x}^\infty - \mathbf{x}^\star\|^2$$

$$= \frac{1}{n}\|\mathbf{x}^\infty - \bar{\mathbf{x}}^\infty + \bar{\mathbf{x}}^\infty - \mathbf{x}^\star\|^2$$

$$\overset{(a)}{=} \frac{1}{n}\|\mathbf{x}^\infty - \bar{\mathbf{x}}^\infty\|^2 + \frac{1}{n}\|\bar{\mathbf{x}}^\infty - \mathbf{x}^\star\|^2$$

$$\geq \frac{1}{n}\|\bar{\mathbf{x}}^\infty - \mathbf{x}^\star\|^2 = \|\bar{x}^\infty - x^\star\|^2 \overset{(48)}{=} B^2, \tag{49}$$

where equality (a) holds because

$$\langle \mathbf{x}^\infty - \bar{\mathbf{x}}^\infty, \bar{\mathbf{x}}^\infty - \mathbf{x}^\star \rangle = (\mathbf{x}^\infty)^T (I - \frac{1}{n}\mathbb{1}\mathbb{1}^T) \cdot \mathbb{1}(\bar{x}^\infty - x^\star) = 0. \tag{50}$$

Inequality (49) implies that the term $B^2$ cannot be removed from bound (39) with any improved analysis for our constructed problem.

### B.4 QG-DADAM AND ADAPT-THEN-COMMUNICATE STRUCTURE

The QG-DAdam algorithm (Lin et al., 2021) is listed in Algorithm 3.

### B.5 FIXED POINT CONDITION FOR QG-DADAM

We let $x_i^{(t)} \to x_i^\infty$ as $t \to \infty$ for any $i \in [n]$. We also assume each node $i$ can access the full-batch gradient $\nabla f_i(x)$ in order to remove the influence of gradient noise. Under such setting, we can verify in QG-DAdam that

$$d_i^\infty = 0, \quad \hat{m}_i^\infty = 0, \quad \hat{v}_i^\infty = 0$$
$$m_i^\infty = (1-\beta_1)\nabla f_i(x_i^\infty), \quad v_i^\infty = (1-\beta_2)\nabla f_i(x_i^\infty) \odot \nabla f_i(x_i^\infty) \tag{51}$$

Substituting $m_i^\infty$ and $v_i^\infty$ to the highlighted step in QG-DAdam, we can achieve the fixed-point condition as follows

$$x_i^\infty = \sum_{j \in \mathcal{N}_i} w_{ij}\big(x_j^\infty - \gamma(1-\beta_1)(\sqrt{1-\beta_2}H_j + \epsilon I)^{-1}\nabla f_j(x_j^\infty)\big), \quad \forall i \in [n]. \tag{52}$$

where $H_i = \mathrm{diag}\{\sqrt{\nabla f_i(x_i^\infty) \odot \nabla f_i(x_i^\infty)}\}$. If we further let $\bar{x}^\infty := \frac{1}{n}\sum_{i=1}^n x_i^\infty$, the following relation can be achieved by (52) and Assumption A.3:

$$\bar{x}^\infty = \bar{x}^\infty - \frac{\gamma(1-\beta_1)}{n}\sum_{i=1}^n (\sqrt{1-\beta_2}H_i + \epsilon I)^{-1}\nabla f_i(x_i^\infty)$$

$$\implies \frac{1}{n}\sum_{i=1}^n (\sqrt{1-\beta_2}H_i + \epsilon I)^{-1}\nabla f_i(x_i^\infty) = 0$$

$$\implies \frac{1}{n}\sum_{i=1}^n (\sqrt{1-\beta_2}H_i + \epsilon I)^{-1}\nabla f_i(x^\infty) = 0 \quad \text{(if } x_1^\infty = \cdots = x_n^\infty = x^\infty \text{ in the limit).} \tag{53}$$

The last relation is the one listed in (10).

### B.6 Proof of Proposition 2

The proof of Proposition 2 is similar to that of Proposition 1. With notations defined in Sec. B.3, we can rewrite (52) into the following compact manner:

$$\mathbf{x}^\infty = \mathcal{W}\big(\mathbf{x}^\infty - \gamma(1-\beta_1)(\sqrt{1-\beta_2}\mathbf{H} + \epsilon I)^{-1}\nabla\mathbf{f}(\mathbf{x}^\infty)\big) \tag{54}$$

**Upper bound of** $\|\bar{x}^\infty - x^\star\|$. Left-multiplying $\frac{1}{n}\mathbb{1}^T$ to both sides of recursion (30), we achieve

$$\frac{1}{n}\sum_{i=1}^n (\sqrt{1-\beta_2}H_i + \epsilon I)^{-1}\nabla f_i(x_i^\infty) = 0. \tag{55}$$

Using a similar argument to that in (32) and (34), we have

$$\sqrt{n}\|\bar{x}^\infty - x^\star\| \leq \frac{2\sqrt{n}B}{\mu} + \frac{2L}{\mu}\|\mathbf{x}^\infty - \bar{\mathbf{x}}^\infty\| \tag{56}$$

in which the constant bias term $B$ is defined as

$$B := \|\frac{1}{n}\sum_{i=1}^n (\sqrt{1-\beta_2}H_i + \epsilon I)^{-1}\nabla f_i(x_i^\infty) - \frac{1}{n}\sum_{i=1}^n \nabla f_i(x_i^\infty)\|. \tag{57}$$

**Upper bound of** $\|\mathbf{x}^\infty - \bar{\mathbf{x}}^\infty\|$. Subtracting $\bar{\mathbf{x}}^\infty$ from both sides of (54), we have

$$(I - \mathcal{W})(\mathbf{x}^\infty - \bar{\mathbf{x}}^\infty) = -\gamma(1-\beta_1)\mathcal{W}(\sqrt{1-\beta_2}\mathbf{H} + \epsilon I)^{-1}\nabla\mathbf{f}(\mathbf{x}^\infty). \tag{58}$$

Using similar arguments as in (36) and (37), and facts that $\|(\sqrt{1-\beta_2}\mathbf{H} + \epsilon I)^{-1}\| = \max_i\{\|(\sqrt{1-\beta_2}H_i + \epsilon I)^{-1}\|_2\} \leq \frac{1}{\epsilon}$ and $\|\mathcal{W}\| \leq 1$, we have

$$\|\mathbf{x}^\infty - \bar{\mathbf{x}}^\infty\| \leq \frac{\gamma(1-\beta_1)\|\nabla\mathbf{f}(\mathbf{x}^\infty)\|}{(1-\rho)\epsilon} \leq \frac{\gamma G}{(1-\rho)\epsilon}. \tag{59}$$

**Upper bound of** $\lim_{t\to\infty}\frac{1}{n}\sum_{i=1}^n \|x_i^{(t)} - x^\star\|^2$. Using similar arguments to those in (38) and (39), we achieve

$$\lim_{t\to\infty}\frac{1}{n}\sum_{i=1}^n \|x_i^{(t)} - x^\star\|^2 = \frac{1}{n}\|\mathbf{x}^\infty - \mathbf{x}^\star\|^2 = O\Big(\frac{\gamma^2 G^2}{(1-\rho)^2\epsilon^2} + B^2\Big). \tag{60}$$

**Lower bound of** $\lim_{t\to\infty}\frac{1}{n}\sum_{i=1}^n \|x_i^{(t)} - x^\star\|^2$. With the same constructed problem as in (41), we can easily establish

$$\lim_{t\to\infty}\frac{1}{n}\sum_{i=1}^n \|x_i^{(t)} - x^\star\|^2 \geq B^2 \tag{61}$$

following the arguments in (42) – (50), where $B$ is defined in (57).

### B.7 GT-DADAM

**Fixed-point condition.** If we define $\bar{H} = \frac{1}{n}\sum_{i=1}^{n} H_i$, the fixed-point condition of GT-DAdam (i.e., Algorithm 2 in (Chen et al., 2021a)) is as follows:

$$x_i^\infty = \sum_{j\in\mathcal{N}_i} w_{ij}x_j^\infty - \gamma(\bar{H} + \epsilon I)^{-1}\nabla f_i(x_i^\infty), \quad \forall i \in [n]. \tag{62}$$

$$\implies \quad \bar{x}^\infty = \bar{x}^\infty - \frac{\gamma}{n}\sum_{i=1}^{n}(\bar{H} + \epsilon I)^{-1}\nabla f_i(x_i^\infty)$$

$$\implies \quad \frac{1}{n}\sum_{i=1}^{n}\nabla f_i(x^\infty) = 0 \quad \text{(If } x_1^\infty = \cdots = x_n^\infty = x^\infty \text{ in the limit)} \tag{63}$$

It is observed that the fixed-point of GT-DAdam satisfies the optimality condition of problem (1).

**Proposition 4** *Under the same conditions as in Proposition 1, the distance between GT-DAdam fixed point and the global solution $x^\star$ can be characterized as follows:*

$$\lim_{t\to\infty}\sum_{i=1}^{n}\|x_i^{(t)} - x^\star\|^2 = O\Big(\frac{\gamma^2 G^2}{(1-\rho)^2\epsilon^2}\Big) \tag{64}$$

*Proof.* With notations defined in Sec. B.3, the fixed-point condition in (62) can be rewritten as

$$\mathbf{x}^\infty = \mathcal{W}\mathbf{x}^\infty - \gamma(\bar{\mathbf{H}} + \epsilon I)^{-1}\nabla\mathbf{f}(\mathbf{x}^\infty) \tag{65}$$

where $\bar{\mathbf{H}} = \text{diag}\{\bar{H}, \cdots, \bar{H}\} \in \mathbb{R}^{nd\times nd}$.

**Bound of $\|\bar{x}^\infty - x^\star\|$.** Left-multiplying $\frac{1}{n}\mathbb{1}^T$ to both sides of recursion (65), we achieve

$$\frac{1}{n}\sum_{i=1}^{n}\nabla f_i(x^\infty) = 0. \tag{66}$$

With the above fact, we have

$$\|\bar{x}^\infty - x^\star\| = \|\bar{x}^\infty - x^\star - \gamma\big(\frac{1}{n}\sum_{i=1}^{n}\nabla f_i(x_i^\infty) - \frac{1}{n}\sum_{i=1}^{n}\nabla f_i(x^\star)\big)\|$$

$$\leq \|\bar{x}^\infty - x^\star - \gamma\big(\frac{1}{n}\sum_{i=1}^{n}\nabla f_i(\bar{x}^\infty) - \frac{1}{n}\sum_{i=1}^{n}\nabla f_i(x^\star)\big)\|$$

$$+ \gamma\|\frac{1}{n}\sum_{i=1}^{n}\nabla f_i(x_i^\infty) - \frac{1}{n}\sum_{i=1}^{n}\nabla f_i(\bar{x}^\infty)\|$$

$$\leq (1 - \frac{\gamma\mu}{2})\|\bar{x}^\infty - x^\star\| + \frac{\gamma L}{\sqrt{n}}\|\mathbf{x}^\infty - \bar{\mathbf{x}}^\infty\| \tag{67}$$

where the last inequality holds since each $f_i(x)$ is $\mu$-strongly convex and $L$-smooth. We thus have

$$\sqrt{n}\|\bar{x}^\infty - x^\star\| \leq \frac{2L}{\mu}\|\mathbf{x}^\infty - \bar{\mathbf{x}}^\infty\| \tag{68}$$

**Bound of $\|\mathbf{x}^\infty - \bar{\mathbf{x}}^\infty\|$.** Subtracting $\bar{\mathbf{x}}^\infty$ from both sides of (65), we have

$$(I - \mathcal{W})(\mathbf{x}^\infty - \bar{\mathbf{x}}^\infty) = -\gamma(\bar{\mathbf{H}} + \epsilon I)^{-1}\nabla\mathbf{f}(\mathbf{x}^\infty). \tag{69}$$

With inequality (43) in (Yuan et al., 2021), we have

$$\|(I - \mathcal{W})(\mathbf{x}^\infty - \bar{\mathbf{x}}^\infty)\| \geq (1-\rho)\|\mathbf{x}^\infty - \bar{\mathbf{x}}^\infty\| \tag{70}$$

With (69) and (70), and noting that $\|(\bar{H} + \epsilon I)^{-1}\|_2 \leq 1/\epsilon$, we achieve

$$\|\mathbf{x}^\infty - \bar{\mathbf{x}}^\infty\| \leq \frac{\gamma\|\nabla\mathbf{f}(\mathbf{x}^\infty)\|}{(1-\rho)\epsilon} \tag{71}$$

**Proof of Proposition 4.** With (68) and (71), we have

$$
\begin{aligned}
\|\mathbf{x}^\infty - \mathbf{x}^\star\| &\le \|\mathbf{x}^\infty - \bar{\mathbf{x}}^\infty\| + \sqrt{n}\|\bar{x}^\infty - x^\star\| \\
&\le (1 + \frac{2L}{\mu})\|\mathbf{x}^\infty - \bar{\mathbf{x}}^\infty\| \\
&\le (1 + \frac{2L}{\mu})\frac{\gamma}{(1-\rho)\epsilon}\|\nabla \mathbf{f}(\mathbf{x}^\infty)\|.
\end{aligned}
\tag{72}
$$

The above inequality implies that

$$
\lim_{t\to\infty} \frac{1}{n}\sum_{i=1}^n \|x_i^{(t)} - x^\star\|^2 = O\Big(\frac{\gamma^2 G^2}{(1-\rho)^2\epsilon^2}\Big).
\tag{73}
$$

∎

**Remark 3** *Different from Propositions 1 and 2, it is observed from Proposition 4 that the distance between the GT-DAdam fixed point and the optimal solution will diminish to zero as learning rate $\gamma \to 0$. However, GT-DAdam will incur two rounds of neighborhood communication per iteration, which is much more expensive than DAdam and QG-DAdam.*

## C  FIXED-POINT CONDITION FOR DAG-ADAM

### C.1  FIXED-POINT CONDITION

When $t \to \infty$, we let $x_i^{(t)} \to x_i^\infty$ for any $i \in [n]$ in DAG-Adam. We further assume each agent can access the real gradient $\nabla f_i(x)$ instead of the stochastic gradient $\nabla F(x;\xi_i)$. Under this setting, we can verify that

$$
\hat{g}_i^\infty = \nabla f_i(x_i^\infty) + \frac{\nu}{\gamma}(x_i^\infty - \sum_{j\in\mathcal{N}_i} w_{ij}x_j^\infty)
\tag{74}
$$

and $m_i^\infty = \hat{g}_i^\infty$, $v_i^\infty = \hat{g}_i^\infty \odot \hat{g}_i^\infty$. By letting $\hat{H}_i = \text{diag}\{\sqrt{v_i^\infty}\}$, it can be derived from the adptive step in DAG-Adam that

$$
x_i^\infty = x_i^\infty - \gamma(\hat{H}_i + \epsilon I)^{-1}\hat{g}_i^\infty, \quad \forall i \in [n]
\tag{75}
$$

$$
\iff \quad \nabla f_i(x_i^\infty) + \frac{\nu}{\gamma}(x_i^\infty - \sum_{j\in\mathcal{N}_i} w_{ij}x_j^\infty) = 0, \quad \forall i \in [n]
\tag{76}
$$

$$
\implies \quad \frac{1}{n}\sum_{i=1}^n \nabla f_i(x_i^\infty) = 0
\tag{77}
$$

$$
\iff \quad \frac{1}{n}\sum_{i=1}^n \nabla f_i(x^\infty) = 0 \quad \text{(If } x_i^\infty = x^\infty \text{ in the limit)}
\tag{78}
$$

In other words, DAG-Adam can converge to the stationary solution if $x_1^\infty = \cdots = x_n^\infty = x^\infty$ in the limit. Compared to DAdam and QG-DAdam in which the discrepancy between $H_i$'s has made the limiting fixed point deviate from the desired solution, the communicate-then-adapt structure in DAG-Adam make it immune to the difference between $H_i$.

### C.2  PROOF OF PROPOSITION 3.

With notations defined in Sec. B.3, the fixed-point condition in (76) can be rewritten as

$$
\nabla \mathbf{f}(\mathbf{x}^\infty) + \frac{\nu}{\gamma}(I - \mathcal{W})\mathbf{x}^\infty = 0.
\tag{79}
$$

**Bound of $\|\bar{x}^\infty - x^\star\|$.** Left-multiplying $\frac{1}{n}\mathbb{1}^T$ to both sides of recursion (79), we achieve

$$
\frac{1}{n}\sum_{i=1}^n \nabla f_i(x^\infty) = 0.
\tag{80}
$$

Table 5: Comparison between different decentralized adaptive methods in the limiting point distance to the global solution $x^\star$ when cost function in problem (1) is strongly convex. The learning rate $\gamma$ is set as constant, and the full-batch gradient $\nabla f_i(x)$ can be assessed per-iteration.

| Algorithm | Distance to global solution |
|---|---|
| DAdam (Nazari et al., 2019) | $\frac{\gamma^2 G^2}{(1-\rho)^2 \epsilon^2} + B^2$ |
| QG-DAdam (Lin et al., 2021) | $\frac{\gamma^2 G^2}{(1-\rho)^2 \epsilon^2} + B^2$ |
| GT-DAdam (Chen et al., 2021a) | $\frac{\gamma^2 G^2}{(1-\rho)^2 \epsilon^2}$ |
| DAG-Adam (ours; $\nu = 1$) | $\frac{\gamma^2 G^2}{(1-\rho)^2}$ |

Using similar arguments to (67), we achieve

$$\sqrt{n}\|\bar{x}^\infty - x^\star\| \leq \frac{2L}{\mu}\|\mathbf{x}^\infty - \bar{\mathbf{x}}^\infty\| \tag{81}$$

**Bound of $\|\mathbf{x}^\infty - \bar{\mathbf{x}}^\infty\|$.** Subtracting $\bar{\mathbf{x}}^\infty$ from both sides of (79), we have

$$(I - \mathcal{W})(\mathbf{x}^\infty - \bar{\mathbf{x}}^\infty) = -\frac{\gamma}{\nu}\nabla\mathbf{f}(\mathbf{x}^\infty). \tag{82}$$

With inequality (43) in (Yuan et al., 2021), we have

$$\|(I - \mathcal{W})(\mathbf{x}^\infty - \bar{\mathbf{x}}^\infty)\| \geq (1 - \rho)\|\mathbf{x}^\infty - \bar{\mathbf{x}}^\infty\| \tag{83}$$

With (82) and (83), we achieve

$$\|\mathbf{x}^\infty - \bar{\mathbf{x}}^\infty\| \leq \frac{\gamma\|\nabla\mathbf{f}(\mathbf{x}^\infty)\|}{(1 - \rho)\nu} \tag{84}$$

**Proof of Proposition 3.** With (81) and (84), we have

$$\begin{aligned}
\|\mathbf{x}^\infty - \mathbf{x}^\star\| &\leq \|\mathbf{x}^\infty - \bar{\mathbf{x}}^\infty\| + \sqrt{n}\|\bar{x}^\infty - x^\star\| \\
&\leq (1 + \frac{2L}{\mu})\|\mathbf{x}^\infty - \bar{\mathbf{x}}^\infty\| \\
&\leq (1 + \frac{2L}{\mu})\frac{\gamma}{(1 - \rho)\nu}\|\nabla\mathbf{f}(\mathbf{x}^\infty)\|.
\end{aligned} \tag{85}$$

The above inequality implies that

$$\lim_{t\to\infty} \frac{1}{n}\sum_{i=1}^{n}\|x_i^{(t)} - x^\star\|^2 = O\Big(\frac{\gamma^2 G^2}{(1 - \rho)^2\nu^2}\Big). \tag{86}$$

### C.3 COMPARISON IN LIMITING POINT DISTANCES TO GLOBAL SOLUTION

Table 5 lists the results we establish in Propositions $1 - 4$. When $\nu = 1$ in the DAG-DAdam method, which is a valid choice for convex settings in empirical studies, the comparison between different decentralized adaptive methods in the limiting point distance to the global solution $x^\star$ can be listed as follows:

$$\text{DAG-Adam} < \text{GT-DAdam} < \text{QG-DAdam} \approx \text{DAdam} \tag{87}$$

This conclusion matches with the numerical experiment in Fig. 1.

### C.4 EXPERIMENT SETTING FOR LOGISTIC REGRESSION (I.E., FIG 1)

In this experiment, we consider the binary logistic regression problem:

$$\min_{x\in\mathbb{R}^d} \frac{1}{n}\sum_{i=1}^{n} f_i(x) + \frac{\lambda}{2}\|x\|^2 \quad \text{where } f_i(x) = \frac{1}{K}\sum_{k=1}^{K} \ln\big(1 + \exp(-\gamma_{i,k}h_{i,k}^T x)\big) \tag{88}$$

where $h_{i,k} \in \mathbb{R}^d$ is the $k$-th feature vector at agent $i$ and $\gamma_{i,k} \in \{+1, -1\}$ is the corresponding label. In the experiment, we set $\lambda = 0.25$, $n = 16$, and all computing nodes are organized into the (undirected) ring topology. The weighted matrix is simply generated through the average rule so that is satifies Assumption A.3. Also, we choose $d = 25$ as the dimension of feature vector and generate $K = 50$ data for each agent locally. The corresponding $\gamma$ is generated through a pre-set random $x^o$ vector. We first compute the linear score $h_{i,k}^T x^o$, then determine the $\gamma$ by the probability based on the sigmoid value. For all algorithms, we choose the constant learning rate $\gamma = 0.01$, $\beta_1 = 0.9$, $\beta_2 = 0.99$. One exception is that we choose $\beta_1 = 0.1$ for QG-DAdam algorithm to achieve better convergence performance.

Besides the learning curve we have shown in Fig. 1, we also plots the $v_i^{(t)}$ of various algorithms at the last iteration in Fig. 5. To characterize the difference between different agents, we use the box plot to show the difference of $v_i^{(t)}$ of each entry among all agents. From the figure, it is clear that the $v_i^{(t)}$ in DAG-Adam algorithm is almost zero when the algorithm converged, which is much narrower and lower than others methods.

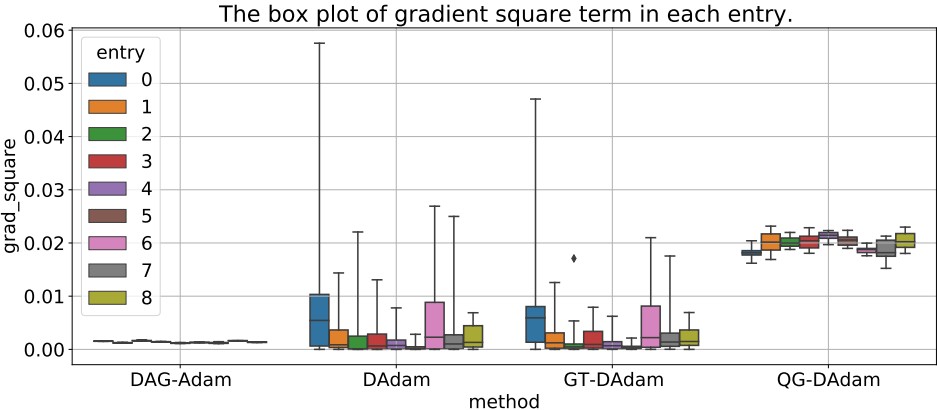

Figure 5: The boxplot of the first 9 entries of gradient square term $v_i^{(t)}$ at the end of iteration of various algorithms. The box represents the mean, quartile, and the min-max value of that entries cross all agents.

Fig. 6 provides complementary experimental results to Fig. 1. We added a new algorithm named DAdam-ReParam which reparameterizes DAdam to obtain a communicate-then-adapt structure. In addition, the convergence with both constant and decaying learning rates are depicted in the left and right plot in Fig. 6, respectively. The It is observed that DAdam-ReParam converges closely to DAdam, and both approaches converge to a less accurate solution than DAG-Adam.

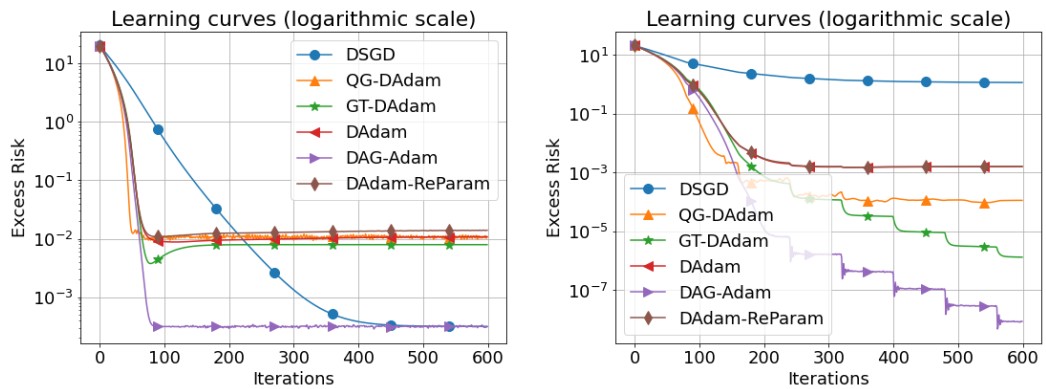

Figure 6: Convergence comparison between various algorithms for a full-batch logistic regression problem. The $y$-axis indicates the excess risk, i.e., $\frac{1}{n} \sum_{i=1}^n \mathbb{E}(f(x_i^{(t)}) - f^\star)$. Left plot: convergence with constant learning rate. Right plot: convergence with stair-wise decaying learning rate.

## D  More discussion on the hyper-parameter $\nu$

This section discusses the role of hyper-parameter $\nu$ in DAG-Adam, and how it influences the convergence performance. To this end, we first discuss whether $\nu$ is needed in the vanilla decentralized SGD (or DSGD for short) algorithm and DAdam. Next, we will study the influence of $\nu$ in the more complicated DAG-Adam algorithm.

### D.1  Influence of $\nu$ on DSGD

Without loss of generality, we consider the DSGD method with a full-batch gradient. In this setting, the augmented gradient of DSGD (13) is given by

$$x_i^{(t+1)} = x_i^{(t)} - \gamma\big(\underbrace{\nabla f_i(x_i^{(t)}) + \frac{1}{\gamma}(x_i^{(t)} - \sum_{j\in\mathcal{N}_i} w_{ij}x_j^{(t)})}_{\text{augmented gradient}}\big), \quad \forall\, i \in [n]. \tag{89}$$

With notations defined in Appendix A and the introduction of the hyper-parameter $\nu$ as in DAG-Adam, we can rewrite the above recursion into a more compact form

$$\mathbf{x}^{(t+1)} = \mathbf{x}^{(t)} - \gamma\Big(\nabla\mathbf{f}(\mathbf{x}^{(t)}) + \frac{\nu}{\gamma}(I - \mathcal{W})\mathbf{x}^{(t)}\Big)$$
$$= \underbrace{\Big((1-\nu)I + \nu\mathcal{W}\Big)}_{:=\widetilde{\mathcal{W}}}\mathbf{x}^{(t)} - \gamma\nabla\mathbf{f}(\mathbf{x}^{(t)}) \tag{90}$$

The above recursion implies that the hyper-parameter $\nu$ will replace the original weight matrix $\mathcal{W}$ with $\widetilde{\mathcal{W}}$, whose eigenvalues are derived as

$$\mathrm{eig}_k(\widetilde{\mathcal{W}}) = (1-\nu)\cdot 1 + \nu\cdot\mathrm{eig}_k(\mathcal{W}) \iff 1 - \mathrm{eig}_k(\widetilde{\mathcal{W}}) = \nu(1 - \mathrm{eig}_k(\mathcal{W})) \tag{91}$$

where $\mathrm{eig}_k(\mathcal{W})$ stands for the $k$-th eigenvalue of matrix $\mathcal{W}$. Equation (91) implies all eigenvalues of $\widetilde{\mathcal{W}}$ are more closer to value 1 for any $\nu \in [0,1)$. Since a larger spectral gap, i.e. $1 - \lambda_2(\widetilde{\mathcal{W}})$, will reduce the mixing time of gossip average and hence speed up the DSGD convergence, the optimal choice for DSGD is $\nu = 1$; other $\nu$ values will not improve the convergence of the vanilla DSGD.

We validate the above discussion with deep learning experiments over CIFAR-10 dataset. Fig. 7 illustrates the influence of $\nu$ on vanilla DSGD. It is observed that $\nu = 1$ enables DSGD to converge fastest and achieve the best validation accuracy, which is consistent with the above conclusions.

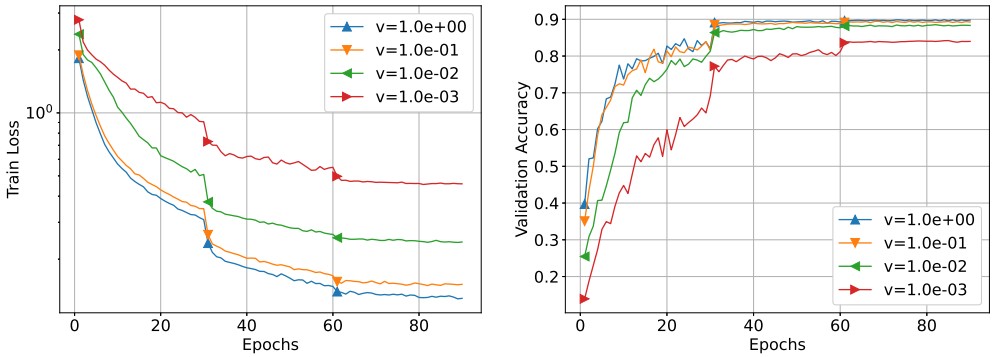

Figure 7: Loss and accuracy evolution for CIFAR-10 dataset using DSGD($\nu$) under a homogeneous network of size 8. Left plot: training loss. Right plot: validation accuracy.

### D.2  Influence of $\nu$ on DAdam

Since DAdam is an immediate variant of DSGD, we expect the influence of $\nu$ on DAdam will be similar to that on DSGD. To validate it, we impose a hyper-parameter $\nu$ to DAdam recursion:

$$\mathbf{g}^{(t)} = \nabla\mathbf{f}(\mathbf{x}^{(t)}) \tag{92}$$

$$\mathbf{m}^{(t)} = \beta_1 \mathbf{m}^{(t-1)} + (1 - \beta_1)\mathbf{g}^{(t)} \tag{93}$$

$$\mathbf{v}^{(t)} = \beta_2 \mathbf{v}^{(t-1)} + (1 - \beta_2)(\mathbf{g}^{(t)} \odot \mathbf{g}^{(t)}) \tag{94}$$

$$\mathbf{x}^{(t+1)} = ((1 - \nu)I + \nu W)\mathbf{x}^{(t)} - \gamma(\mathbf{H}^{(t)} + \epsilon I)^{-1}\mathbf{m}^{(t)} \tag{95}$$

Fig. 8 illustrates the influence of $\nu$ on DAdam. Similar to DSGD, it is observed that DAdam with $\nu = 1$ (i.e., the vanilla DAdam with no introduction of $\nu$) has the best convergence performance.

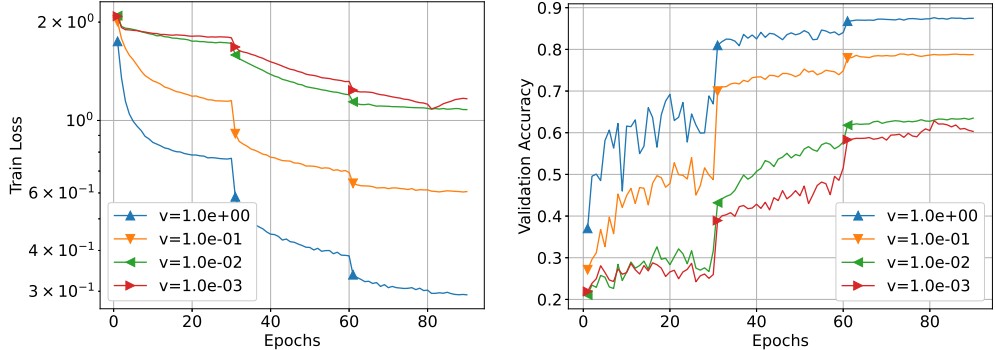

Figure 8: Loss and accuracy evolution for CIFAR-10 dataset using DAdam($\nu$) under a homogeneous network of size 8. Left plot: training loss. Right plot: validation accuracy.

### D.3 INFLUENCE OF $\nu$ ON DAG-ADAM

Without loss of generality, we consider DAG-Adam with dimension $d = 1$. In this scenario, DAG-Adam can be rewritten as

$$\mathbf{g}^{(t)} = \nabla \mathbf{f}(\mathbf{x}^{(t)}) + \frac{\nu}{\gamma}(I - W)\mathbf{x}^{(t)} \tag{96}$$

$$\mathbf{m}^{(t)} = \beta_1 \mathbf{m}^{(t-1)} + (1 - \beta_1)\mathbf{g}^{(t)} \tag{97}$$

$$\mathbf{v}^{(t)} = \beta_2 \mathbf{v}^{(t-1)} + (1 - \beta_2)(\mathbf{g}^{(t)} \odot \mathbf{g}^{(t)}) \tag{98}$$

$$\mathbf{x}^{(t+1)} = \mathbf{x}^{(t)} - \gamma(\mathbf{H}^{(t)} + \epsilon I)^{-1}\mathbf{m}^{(t)} \tag{99}$$

where $W \in \mathbb{R}^{n \times n}$ is the weight matrix and $\mathbf{H}^{(t)} = \text{diag}\{\sqrt{\mathbf{v}^{(t)}}\} \in \mathbb{R}^{n \times n}$ is a diagonal matrix. Subtracting (99) by $\beta_1$ times (99) with one iteration shift, we have

$$\mathbf{x}^{(t+1)} - \beta_1 \mathbf{x}^{(t)} = \mathbf{x}^{(t)} - \beta_1 \mathbf{x}^{(t-1)} - \gamma(\mathbf{H}^{(t)} + \epsilon I)^{-1}\mathbf{m}^{(t)} + \beta_1\gamma(\mathbf{H}^{(t-1)} + \epsilon I)^{-1}\mathbf{m}^{(t-1)} \tag{100}$$

Rearranging the terms in the above recursion, we obtain

$$\mathbf{x}^{(t+1)} = \mathbf{x}^{(t)} - (1 - \beta_1)\gamma(\mathbf{H}^{(t)} + \epsilon)^{-1}\mathbf{g}^{(t)} + \beta_1(\mathbf{x}^{(t)} - \mathbf{x}^{(t-1)})$$
$$+ \beta_1\Big(\gamma(\mathbf{H}^{(t-1)} + \epsilon)^{-1} - \gamma(\mathbf{H}^{(t)} + \epsilon)^{-1}\Big)\mathbf{m}^{(t-1)} \tag{101}$$

In the following, we will first study a simple example that can shed lights on the influence of $\nu$ on DAG-Adam, and next we will discuss the more generalized scenario.

**An illustrating example.** We consider a trivial scenario in which $\mathbf{H}^{(t)} = \alpha I$, which amounts to the case where all nodes maintain constant $v_i^{(t)}$ across $i$ and $t$ in the local buffer. As we have shown in the convex experiment in Fig. 5, this trivial scenario becomes valid in the asymptotic stage of DAG-Adam in which $v_i^{(t)}$ converges to the same value $v$ for any $i \in [n]$.

Substituting $\mathbf{H}^{(t)} = \alpha I$ into (101), we have

$$\mathbf{x}^{(t+1)} = \mathbf{x}^{(t)} - \frac{\gamma(1 - \beta_1)}{\alpha + \epsilon}\mathbf{g}^{(t)} + \beta_1(\mathbf{x}^{(t)} - \mathbf{x}^{(t-1)})$$
$$= \underbrace{\Big[\big(1 - \frac{1 - \beta_1}{\nu(\alpha + \epsilon)}\big)I + \frac{1 - \beta_1}{\nu(\alpha + \epsilon)}W\Big]}_{:=\widetilde{W}_{\text{DAG}}}\mathbf{x}^{(t)} - \frac{\gamma(1 - \beta_1)}{\alpha + \epsilon}\nabla\mathbf{f}(\mathbf{x}^{(t)}) + \beta_1(\mathbf{x}^{(t)} - \mathbf{x}^{(t-1)})$$

$$\tag{102}$$

From the expression of $\widetilde{W}_{\text{DAG}}$, it is observed that the optimal value for $\nu$ to minimize the mixing time of the weight matrix is $\frac{\alpha+\epsilon}{1-\beta_1}$, which is not $1$ as in the vanilla DSGD algorithm. In addition, it is worth noting that $\widetilde{W}_{\text{DAG}}$ cannot outperform $W$ in terms of mixing time for any choice of $\nu$.

This trivial example illustrates the necessity to have the hyper-parameter $\nu$ in DAG-Adam. The introduction of $\nu$ is to compensate the negative influence of the communicate-then-adapt structure in DAG-Adam which incorporates both the gradient and the consensus constraint into the adaptive momentum buffer. The weight matrix $\widetilde{W}_{\text{DAG}}$ can be quite inefficient to aggregate information if no $\nu$ exists to speed up partial averaging.

**Generalized scenario.** Unlike the above trivial scenario, it is very challenging to determine a proper value for $\nu$ in the general scenarios. To see it, we ignore the last term in (101) to achieve

$$\mathbf{x}^{(t+1)} \approx \underbrace{\left(I - \nu(1-\beta_1)(\mathbf{H}^{(t)} + \epsilon I)^{-1}(I - W)\right)}_{:=\widetilde{W}_{\text{DAG}}} \mathbf{x}^{(t)} - (1-\beta_1)\gamma(\mathbf{H}^{(t)} + \epsilon I)^{-1}\nabla\mathbf{f}(\mathbf{x}^{(t)})$$
$$+ \beta_1(\mathbf{x}^{(t)} - \mathbf{x}^{(t-1)}) \tag{103}$$

which is a reasonable approximation to (101) since $\beta_2$ is typically set as $0.999$ and $\mathbf{v}^{(t)}$ (and hence $\mathbf{H}^{(t)}$) will not change dramatically. The first term in (103) mixes $x_i^{(t)}$ with weight matrix

$$\widetilde{W}_{\text{DAG}} = (I - \nu(1-\beta_1)(\mathbf{H}^{(t)} + \epsilon I)^{-1})I + \nu(1-\beta_1)(\mathbf{H}^{(t)} + \epsilon I)^{-1}W \tag{104}$$

If we let $\Lambda := \nu(1-\beta_1)(\mathbf{H}^{(t)} + \epsilon)^{-1}$, it holds that

$$\Lambda^{-1/2}\widetilde{W}_{\text{DAG}}\Lambda^{1/2} = I - \Lambda + \Lambda^{1/2}W\Lambda^{1/2} \tag{105}$$

The left-hand side of (105) is the similarity transformation of $\widetilde{W}_{\text{DAG}}$. Hence, the eigenvalues of $\widetilde{W}_{\text{DAG}}$ equal to ones of the right-hand side of (105). Unfortunately, it is difficult to determine the eigenvalues of $\Lambda^{1/2}W\Lambda^{1/2}$, not to mention the optimal value for $\nu$. However, it is obvious that $\nu = 1$ is not necessarily optimal for DAG-Adam. This motivates us to tune $\nu$ in experiments to boost the convergence performance of DAG-Adam. In fact, the results in Table 1 (with CIFAR-10 dataset), Table 3 (with ImageNet dataset), and Table 4 (with SQuAD dataset) corroborate that $\nu = 1$ is not the optimal choice for DAG-Adam. How to find the optimal $\nu$ automatically is the main direction of our future work.

## E  CONVERGENCE ANALYSIS

### E.1  CONVERGENCE PROOF OF DAG-ADAM

As we motivated in the Section 3, we consider the augmented loss function

$$\mathcal{L}(\mathbf{x}) = \mathbf{f}(\mathbf{x}) + \frac{\nu}{2\gamma}\|\mathbf{x}\|_{I-\mathcal{W}}^2, \quad \text{where } \mathbf{f}(\mathbf{x}) := \sum_{k=1}^{K} f_k(\mathbf{x}_k) \tag{106}$$

It is easy to verify that

$$\nabla_{\mathbf{x}}\mathcal{L}(\mathbf{x}) = \begin{bmatrix} \nabla_{\mathbf{x}_1} f_1(\boldsymbol{x}_1) \\ \nabla_{\mathbf{x}_2} f_2(\boldsymbol{x}_2) \\ \vdots \\ \nabla_{\mathbf{x}_K} f_K(\boldsymbol{x}_K) \end{bmatrix} + \frac{\nu}{\gamma}(I - \mathcal{W})\mathbf{x} = \nabla\mathbf{f}(\mathbf{x}) + \frac{\nu}{\gamma}(I - \mathcal{W})\mathbf{x} \tag{107}$$

which is exactly the same format as our decentralized augmented gradient in equation 20.

**Proof idea.** Therefore, we can view our DAG-Adam over multiple agents as the standard Adam for virtual single agent with a long stacked vector. This observation leads to the following convergence analysis idea. First, we use the similar proof technique as Zaheer et al. (2018) did for the standard Adam to show the convergence of the decentralized augmented gradient. Next, based on that, we establish the consensus lemma to show that the models between all agents are close. Last,

combining two, we finally arrive the convergence of centralized iterate, $\|\nabla f(\bar{x}^{(t)})\|$ as the common decentralized algorithm theorem stated.

It is common in the proof of adaptive gradient methods, for example Zaheer et al. (2018); Chen et al. (2021a; 2018b); Reddi et al. (2019; 2020), to assume the gradient$\nabla f_i(x_i^{(t)})$ and/or stochastic gradient $\nabla F_i(x_i^{(t)}; \xi_i)$ have bounded $\ell_2$ or $\ell_\infty$ norms at all iterations $t$, i.e., $\|\nabla f_i(x_i^{(t)}))\| \leq G$ and $\|\nabla F_i(x_i^{(t)}; \xi_i)\| \leq G$. Hence, it is nature to see we need the similar bounded gradient assumption on the augmented gradient. However, notice our augmented gradient has a $(I - \mathcal{W})\mathbf{x}$ term. So we need to further assume that $\|\mathbf{x}^{(t)}\| \leq C$. This is a relatively strong assumption, but it can be strictly achieved by simply taking a projection step. Also, in the practice, we always observe that the terms $\mathbf{g}^{(t)}$ converges to a very small value without any requirement on the boundary on the iterate, as shown in Fig. 5. In this case, we can show that

$$\|\mathbf{g}^{(t)}\|_\infty \leq \max_i \|\nabla \hat{f}_i(x_i^{(t)})\| + \|\frac{\nu}{\gamma}(I - \mathcal{W})\mathbf{x}^{(t)}\| \leq G + \frac{\nu}{\gamma}\|\mathbf{x}^{(t)}\| \leq \underbrace{G + \kappa C}_{\triangleq G'} \tag{108}$$

Same as the paper Zaheer et al. (2018) did, we show the case that there is no-momentum for the sake of simplicity, i.e., $\beta_1 = 0$, which is typically referred as RMSprop.

**Lemma 1 (Convergence lemma of augmented gradient)** *Suppose the parameters $\gamma, \nu$ and $\beta_2$ satisfy the conditions that $\gamma \leq \frac{\epsilon - 2\nu}{2L}$, $\nu < \min\{\epsilon/2, \kappa\gamma\}$, ($\kappa$ is any positive constant) and $1 - \beta_2 \leq \frac{\epsilon^2}{16(G + \kappa C)^2}$. Under the Assumption A.1 - A.4 and bounded augmented gradient, the augmented gradient generated using DAG-Adam has the following bound*

$$\frac{1}{T}\sum_{t=0}^{T-1} \mathbb{E}\|\nabla \mathbf{f}(\mathbf{x}^{(t)}) + \frac{\nu}{\gamma}(I - \mathcal{W})\mathbf{x}^{(t)}\|^2 \leq O\Big(\frac{\mathcal{L}(\mathbf{x}^{(0)}) - \mathcal{L}^*}{\gamma T}\Big) + O(\sigma^2) \tag{109}$$

*where $\mathcal{L}^*$ is the minimum value of the augmented loss function (106).*

**Proof.** To shorten the notation, we denote the stochastic augmented gradient as

$$\nabla_{\mathbf{x}}\widehat{\mathcal{L}}(\mathbf{x}) \triangleq \nabla\widehat{\mathbf{f}}(\mathbf{x}^{(t)}) + \frac{\nu}{\gamma}(I - \mathcal{W})\mathbf{x}^{(t)} \tag{110}$$

Utilizing the $L$-smooth property and noting that the augmented loss function $\mathcal{L}$ has a new Lipschitz constant $L' = L + \frac{\nu}{\gamma}$, we have

$$\begin{aligned}
\mathcal{L}(\mathbf{x}^{(t+1)}) \leq & \mathcal{L}(\mathbf{x}^{(t)}) + \langle \nabla\mathcal{L}(\mathbf{x}^{(t)}), \mathbf{x}^{(t+1)} - \mathbf{x}^{(t)} \rangle + \frac{L'}{2}\|\mathbf{x}^{(t+1)} - \mathbf{x}^{(t)}\|^2 \\
\leq & \mathcal{L}(\mathbf{x}^{(t)}) - \gamma\Big\langle \nabla_{\mathbf{x}}\mathcal{L}(\mathbf{x}), (\mathbf{H}^{(t)} + \epsilon)^{-1}\big(\nabla\widehat{\mathbf{f}}(\mathbf{x}^{(t)}) + \frac{\nu}{\gamma}(I - \mathcal{W})\mathbf{x}^{(t)}\big) \Big\rangle \\
& + \frac{\gamma^2 L'}{2}\|(\mathbf{H}^{(t)} + \epsilon)^{-1}(\nabla\widehat{\mathbf{f}}(\mathbf{x}^{(t)}) + \frac{\nu}{\gamma}(I - \mathcal{W})\mathbf{x}^{(t)})\|^2 \\
= & \mathcal{L}(\mathbf{x}^{(t)}) - \gamma\Big\langle \nabla_{\mathbf{x}}\mathcal{L}(\mathbf{x}), (\sqrt{\beta_2}\mathbf{H}^{(t-1)} + \epsilon)^{-1}\nabla_{\mathbf{x}}\widehat{\mathcal{L}}(\mathbf{x}) \Big\rangle \\
& + \gamma\Big\langle \nabla_{\mathbf{x}}\mathcal{L}(\mathbf{x}), \big((\sqrt{\beta_2}\mathbf{H}^{(t-1)} + \epsilon)^{-1} - (\mathbf{H}^{(t)} + \epsilon)^{-1}\big)\nabla_{\mathbf{x}}\widehat{\mathcal{L}}(\mathbf{x}) \Big\rangle \\
& + \frac{\gamma^2 L'}{2}\|(\mathbf{H}^{(t)} + \epsilon)^{-1}\nabla_{\mathbf{x}}\widehat{\mathcal{L}}(\mathbf{x})\|^2 \tag{111}
\end{aligned}$$

where the last equality we add and subtract the inner product with $(\sqrt{\beta_2}\mathbf{H}^{(t-1)} + \epsilon)^{-1}\widehat{\mathcal{L}}(\mathbf{x})$. The main reason to do this is observing that $\mathbf{H}^{(t)}$ is correlated with the stochastic gradient $\nabla\widehat{\mathcal{L}}(\mathbf{x})$.

Taking the conditional expectation $\mathbb{E}_t$ of $\mathcal{L}(\mathbf{x}^{(t+1)})$, more precisely, taking the expectation given all previous recursion information $\{\mathbf{x}^{(j)}, \mathbf{v}^{(j)}\}_{j=1}^t$, we obtain

$$\begin{aligned}
\mathbb{E}_t\mathcal{L}(\mathbf{x}^{(t+1)}) \leq & \mathcal{L}(\mathbf{x}^{(t)}) - \gamma\|\nabla_{\mathbf{x}}\mathcal{L}(\mathbf{x})\|^2_{(\sqrt{\beta_2}\mathbf{H}^{(t-1)} + \epsilon)^{-1}} \\
& + \gamma\mathbb{E}_t\Big\langle \nabla_{\mathbf{x}}\mathcal{L}(\mathbf{x}), \big((\sqrt{\beta_2}\mathbf{H}^{(t-1)} + \epsilon)^{-1} - (\mathbf{H}^{(t)} + \epsilon)^{-1}\big)\nabla_{\mathbf{x}}\widehat{\mathcal{L}}(\mathbf{x}) \Big\rangle
\end{aligned}$$

$$+ \frac{\gamma^2 L'}{2} \mathbb{E}_t \|(\mathbf{H}^{(t)} + \epsilon)^{-1} \nabla_\mathbf{x} \widehat{\mathcal{L}}(\mathbf{x})\|^2 \tag{112}$$

Here we can remove the gradient noise by conditional expectation since the weighted diagonal matrix $\sqrt{\beta_2}\mathbf{H}^{(t-1)}$ is deterministic given the all information before iteration $t$ and the zero gradient-noise assumption A.2.

To bound the term $\mathbb{E}_t \|(\mathbf{H}^{(t)} + \epsilon)^{-1} \nabla_\mathbf{x} \widehat{\mathcal{L}}(\mathbf{x})\|^2$, we notice that any diagonal element of weighting matrix satisfies that

$$\left(G + \kappa C + \epsilon\right)^{-1} \leq [(\mathbf{H}^{(t)} + \epsilon)^{-1}]_{i,i} \leq 1/\epsilon, \quad \forall i \tag{113}$$

Apply the inequality (25), we establish that

$$\mathbb{E}_t \|(\mathbf{H}^{(t)} + \epsilon)^{-1} \nabla_\mathbf{x} \widehat{\mathcal{L}}(\mathbf{x})\|^2 \leq \frac{1}{\epsilon} \mathbb{E}_t \|\nabla_\mathbf{x} \widehat{\mathcal{L}}(\mathbf{x})\|^2_{(\mathbf{H}^{(t)}+\epsilon)^{-1}}$$
$$\leq \frac{1}{\epsilon} \mathbb{E}_t \|\nabla_\mathbf{x} \widehat{\mathcal{L}}(\mathbf{x})\|^2_{(\sqrt{\beta_2}\mathbf{H}^{(t-1)}+\epsilon)^{-1}} \tag{114}$$

where the second inequality is because that $\sqrt{\beta_2}\mathbf{H}^{(t-1)} \leq \mathbf{H}^{(t)}$. Next, we bound the cross terms:

$$\mathbb{E}_t \left\langle \nabla_\mathbf{x} \mathcal{L}(\mathbf{x}), \left((\sqrt{\beta_2}\mathbf{H}^{(t-1)} + \epsilon)^{-1} - (\mathbf{H}^{(t)} + \epsilon)^{-1}\right) \nabla_\mathbf{x} \widehat{\mathcal{L}}(\mathbf{x}) \right\rangle$$
$$= \left\langle \nabla_\mathbf{x} \mathcal{L}(\mathbf{x}), \mathbb{E}_t \left((\sqrt{\beta_2}\mathbf{H}^{(t-1)} + \epsilon)^{-1} - (\mathbf{H}^{(t)} + \epsilon)^{-1}\right) \nabla_\mathbf{x} \widehat{\mathcal{L}}(\mathbf{x}) \right\rangle$$
$$\leq \|\nabla_\mathbf{x} \mathcal{L}(\mathbf{x})\|_\infty \|\mathbb{E}_t \left((\sqrt{\beta_2}\mathbf{H}^{(t-1)} + \epsilon)^{-1} - (\mathbf{H}^{(t)} + \epsilon)^{-1}\right) \nabla_\mathbf{x} \widehat{\mathcal{L}}(\mathbf{x})\|_1 \tag{115}$$

where the last inequality is Holder's inequality. Using the same methods as in (Zaheer et al., 2018, Appendix A), we can show that

$$\|\mathbb{E}_t \left((\sqrt{\beta_2}\mathbf{H}^{(t-1)} + \epsilon)^{-1} - (\mathbf{H}^{(t)} + \epsilon)^{-1}\right) \nabla_\mathbf{x} \widehat{\mathcal{L}}(\mathbf{x})\|_1 \leq \frac{\sqrt{1-\beta_2}}{\epsilon} \mathbb{E}_t \|\nabla_\mathbf{x} \widehat{\mathcal{L}}(\mathbf{x})\|^2_{(\sqrt{\beta_2}\mathbf{H}^{(t-1)}+\epsilon)^{-1}} \tag{116}$$

Plugging back, we establish

$$\mathbb{E}_t \left\langle \nabla_\mathbf{x} \mathcal{L}(\mathbf{x}), \left((\sqrt{\beta_2}\mathbf{H}^{(t-1)} + \epsilon)^{-1} - (\mathbf{H}^{(t)} + \epsilon)^{-1}\right) \nabla_\mathbf{x} \widehat{\mathcal{L}}(\mathbf{x}) \right\rangle$$
$$\leq \frac{G'\sqrt{1-\beta_2}}{\epsilon} \mathbb{E}_t \|\nabla_\mathbf{x} \widehat{\mathcal{L}}(\mathbf{x})\|^2_{(\sqrt{\beta_2}\mathbf{H}^{(t-1)}+\epsilon)^{-1}} \tag{117}$$

Substituting (114) and (117) into (112), we obtain

$$\mathbb{E}_t \mathcal{L}(\mathbf{x}^{(t+1)}) \leq \mathcal{L}(\mathbf{x}^{(t)}) - \gamma \|\nabla_\mathbf{x} \mathcal{L}(\mathbf{x})\|^2_{(\sqrt{\beta_2}\mathbf{H}^{(t-1)}+\epsilon)^{-1}}$$
$$+ \left(\frac{\gamma\sqrt{1-\beta_2}G'}{\epsilon} + \frac{\gamma^2 L'}{2\epsilon}\right) \mathbb{E}_t \|\nabla_\mathbf{x} \widehat{\mathcal{L}}(\mathbf{x})\|^2_{(\sqrt{\beta_2}\mathbf{H}^{(t-1)}+\epsilon)^{-1}} \tag{118}$$

Next, we bound the weighted norm of stochastic noisy gradient:

$$\mathbb{E}_t \|\nabla_\mathbf{x} \widehat{\mathcal{L}}(\mathbf{x})\|^2_{(\sqrt{\beta_2}\mathbf{H}^{(t-1)}+\epsilon)^{-1}} = \|\nabla_\mathbf{x} \mathcal{L}(\mathbf{x})\|^2_{(\sqrt{\beta_2}\mathbf{H}^{(t-1)}+\epsilon)^{-1}} + \mathbb{E}_t \|\nabla \mathbf{f}(\mathbf{x}^{(t)}) - \nabla \widehat{\mathbf{f}}(\mathbf{x}^{(t)})\|^2_{(\sqrt{\beta_2}\mathbf{H}^{(t-1)}+\epsilon)^{-1}}$$
$$\leq \|\nabla_\mathbf{x} \mathcal{L}(\mathbf{x})\|^2_{(\sqrt{\beta_2}\mathbf{H}^{(t-1)}+\epsilon)^{-1}} + \frac{1}{\epsilon} \mathbb{E}_t \|\nabla \mathbf{f}(\mathbf{x}^{(t)}) - \nabla \widehat{\mathbf{f}}(\mathbf{x}^{(t)})\|^2$$
$$= \|\nabla_\mathbf{x} \mathcal{L}(\mathbf{x})\|^2_{(\sqrt{\beta_2}\mathbf{H}^{(t-1)}+\epsilon)^{-1}} + \frac{1}{\epsilon} \sum_{i=1}^n \mathbb{E}_t \|\nabla f_i(\mathbf{x}_i^{(t)}) - \nabla \hat{f}_i(\mathbf{x}_i^{(t)})\|^2$$
$$\leq \|\nabla_\mathbf{x} \mathcal{L}(\mathbf{x})\|^2_{(\sqrt{\beta_2}\mathbf{H}^{(t-1)}+\epsilon)^{-1}} + \frac{1}{\epsilon} n\sigma^2 \tag{119}$$

where the first equality is because of independent and zero-mean gradient noise assumption and the first inequality exploited the equation 25. Plugging back and rearranging the terms, we get

$$\gamma \left(1 - \frac{\sqrt{1-\beta_2}G'}{\epsilon} - \frac{\gamma L'}{2\epsilon}\right) \mathbb{E} \|\nabla \mathcal{L}(\mathbf{x}^{(t)})\|^2_{(\sqrt{\beta_2}\mathbf{H}^{(t-1)}+\epsilon)^{-1}} \tag{120}$$

$$\leq \mathbb{E}\,\mathcal{L}(\mathbf{x}^{(t)}) - \mathbb{E}\,\mathcal{L}(\mathbf{x}^{(t+1)}) + \left(\frac{\gamma\sqrt{1-\beta_2}G'}{\epsilon^2} + \frac{\gamma^2 L'}{2\epsilon^2}\right)n\sigma^2 \tag{121}$$

To make the coefficient in the left-hand side of inequality positive, we require that

$$\frac{\gamma L'}{2\epsilon} \leq \frac{1}{4} \quad\Longrightarrow\quad \gamma \leq \frac{\epsilon - 2\nu}{2L}$$

$$\frac{\sqrt{1-\beta_2}G'}{\epsilon} \leq \frac{1}{4} \quad\Longrightarrow\quad 1-\beta_2 \leq \frac{\epsilon^2}{16(G+\kappa C)^2} \tag{122}$$

Hence, we conclude that by taking the telescoping sum:

$$\frac{1}{T}\sum_{t=0}^{T-1}\mathbb{E}\,\|\nabla\mathbf{f}(\mathbf{x}^{(t)}) + \frac{\nu}{\gamma}(I-\mathcal{W})\mathbf{x}^{(t)}\|^2$$

$$\leq 2(\sqrt{\beta_2}G' + \epsilon)\left(\frac{\mathcal{L}(\mathbf{x}^{(0)}) - \mathcal{L}^*}{\gamma T} + \left(\frac{\sqrt{1-\beta_2}G'}{\epsilon^2} + \frac{\gamma L'}{2\epsilon^2}\right)n\sigma^2\right) \tag{123}$$

where we use the fact that $\mathcal{L}^* \leq \mathcal{L}(\mathbf{x})$ for any $\mathbf{x}$. ∎

Above lemma shows that that local augmented gradient will converge to the stationary point within the constant factor of $O(\sigma^2)$ for constant learning rate $\gamma$, which is the same as stated in Theorem 1 of Zaheer et al. (2018). Next, we evaluate that how close the model between each agent.

**Lemma 2 (Convergence of consensus value)** *Under the same condition as Lemma 1, it holds that*

$$\frac{1}{T}\sum_{t=0}^{T-1}\mathbb{E}\,\|\mathbf{x}^{(t)} - \bar{\mathbf{x}}^{(t)}\|^2 = O\left(\frac{\gamma^2}{(1-\rho)^2\nu^2}\right) \tag{124}$$

*where the $\rho$ is the second largest eigenvalue of combination matrix $W$ in magnitude, which is strictly less than 1.*

**Proof**: Since $W$ is a symmetric doubly stochastic matrix, there is an eigenvalue decomposition form $W = Q\Lambda Q^T$, where $Q$ is some orthogonal matrix. The diagonal matrix $\Lambda$ has the value within the range $(0,1]$ and contains one and only one entry that is 1 due to Assumption A.3. Therefore, we can establish that

$$\|(I-\mathcal{W})\mathbf{x}\|^2 = \|(I-\mathcal{W})(\mathbf{x}-\bar{\mathbf{x}})\|^2 = \|(Q(I-\Lambda)Q^T \otimes I)(\mathbf{x}-\bar{\mathbf{x}})\|^2 \geq (1-\rho)^2\|\mathbf{x}-\bar{\mathbf{x}}\|^2 \tag{125}$$

where the first equality holds because $(I-W)\mathbb{1} = 0$. Now, we are read to establish the bound for the consensus error from the augmented gradient:

$$\frac{1}{T}\sum_{t=0}^{T-1}\|\mathbf{x}^{(t)} - \bar{\mathbf{x}}^{(t)}\|^2 \leq \frac{\gamma^2}{(1-\rho)^2\nu^2 T}\sum_{t=0}^{T-1}\left\|\frac{\nu}{\gamma}(I-\mathcal{W})\mathbf{x}^{(t)}\right\|^2$$

$$\leq \frac{2\gamma^2}{(1-\rho)^2\nu^2 T}\sum_{t=0}^{T-1}\left\|\nabla\mathbf{f}(\bar{\mathbf{x}}) + \frac{\nu}{\gamma}(I-\mathcal{W})\mathbf{x}^{(t)}\right\|^2 + \frac{2\gamma^2}{(1-\rho)^2\nu^2 T}\sum_{t=0}^{T-1}\left\|\nabla\mathbf{f}(\bar{\mathbf{x}})\right\|^2 \tag{126}$$

where the last inequality utilizes the Jensen's inequality $\|a\|^2 = \|a+b+(-b)\|^2 \leq 2\|a+b\|^2 + 2\|b\|^2$ and by setting $a = \frac{\nu}{\gamma}(I-\mathcal{W})\mathbf{x}^{(t)}$ and $b = \nabla\mathbf{f}(\mathbf{x}^{(t)})$. Using the conclusion from Lemma 1 and bounded gradient assumption, we can immediately establish that

$$\frac{1}{T}\sum_{t=0}^{T-1}\|\mathbf{x}^{(t)} - \bar{\mathbf{x}}^{(t)}\|^2$$

$$\leq \frac{4\gamma^2(\sqrt{\beta_2}G' + \epsilon)}{(1-\rho)^2\nu^2 T}\left(\frac{\mathcal{L}(\mathbf{x}^{(0)}) - \mathcal{L}^*}{\gamma T} + \left(\frac{\sqrt{1-\beta_2}G'}{\epsilon^2} + \frac{\gamma L'}{2\epsilon^2}\right)n\sigma^2 + nG^2\right) \tag{127}$$

which gives the desired bound that the consensus error has the order of $O\left(\frac{\gamma^2}{(1-\rho^2)\nu^2}\right)$.

■

This bounds for the consensus error has several easy interpretable conclusions: To have the smaller consensus error, we can either 1) use a better connectivity of the graph (i.e., $\rho$ is closer to 0), or 2) set a smaller learning rate $\gamma$, or 3) choose a larger $\nu$. More accurately, the consensus error is determined by the ratio between $\gamma$ and $\nu$, which is straight-forward to understand since the communication term $(I - \mathcal{W})\mathbf{x}$ is scaled by $\nu/\gamma$.

With above two lemma, we are ready to present the convergence theorem.

**Proof of theorem 1**. The first half of statement has already been shown in the lemma 1. We just to show the convergence for the centralized iterate. To connect the gradient over the centralized iterate $\bar{x}$ with the stacked gradient over the stacked iterate $\bar{\mathbf{x}}$, we have

$$
\begin{aligned}
\|\nabla f(\bar{x}^{(t)})\|^2 &\overset{(a)}{=} \left\| \frac{1}{n} \sum_{i=1}^n \nabla f_i(\bar{x}^{(t)}) + \frac{1}{n} \sum_{i=1}^n \frac{\nu}{\gamma}(x_i^{(t)} - \sum_{j \in \mathcal{N}_i} w_{ij} x_j^{(t)}) \right\|^2 \\
&\overset{(b)}{\leq} \frac{1}{n} \sum_{i=1}^n \left\| \nabla f_i(\bar{x}^{(t)}) + \frac{\nu}{\gamma}(x_i^{(t)} - \sum_{j \in \mathcal{N}_i} w_{ij} x_j^{(t)}) \right\|^2 \\
&= \frac{1}{n} \left\| \nabla \mathbf{f}(\bar{\mathbf{x}}^{(t)}) + \frac{\nu}{\gamma}(I - \mathcal{W})\mathbf{x}^{(t)} \right\|^2 \\
&\overset{(c)}{\leq} \frac{2}{n} \left\| \nabla \mathbf{f}(\mathbf{x}^{(t)}) + \frac{\nu}{\gamma}(I - \mathcal{W})\mathbf{x}^{(t)} \right\|^2 + \frac{2}{n} \left\| \nabla \mathbf{f}(\mathbf{x}^{(t)}) - \nabla \mathbf{f}(\mathbf{x}^{(t)}) \right\|^2 \\
&\overset{(d)}{\leq} \frac{2}{n} \left\| \nabla \mathbf{f}(\mathbf{x}^{(t)}) + \frac{\nu}{\gamma}(I - \mathcal{W})\mathbf{x}^{(t)} \right\|^2 + \frac{2L^2}{n} \left\| \mathbf{x}^{(t)} - \mathbf{x}^{(t)} \right\|^2
\end{aligned}
\tag{128}
$$

where step (a) follows the fact that $W = [w_{ij}]$ is a doubly stochastic matrix and step (b) and (c) are due to Jensen's inequality and step(d) utilized the Lipschitz property.

Therefor, combining with the conclusions in lemma 1 and lemma 2, we have

$$
\begin{aligned}
\frac{1}{T} \sum_{t=1}^T &\|\nabla f(\bar{x}^{(t)})\|^2 \\
\leq & \frac{2}{nT} \sum_{t=1}^T \left\| \nabla \mathbf{f}(\mathbf{x}^{(t)}) + \frac{\nu}{\gamma}(I - \mathcal{W})\mathbf{x}^{(t)} \right\|^2 + \frac{2L^2}{nT} \sum_{t=1}^T \left\| \mathbf{x}^{(t)} - \mathbf{x}^{(t)} \right\|^2 \\
\leq & 4(\sqrt{\beta_2}G' + \epsilon)\left( \frac{\mathcal{L}(\mathbf{x}^{(0)}) - \mathcal{L}^*}{\gamma nT} + \left( \frac{\sqrt{1 - \beta_2}G'}{\epsilon^2} + \frac{\gamma L'}{2\epsilon^2} \right)\sigma^2 \right) \\
& + \frac{8L^2\gamma^2(\sqrt{\beta_2}G' + \epsilon)}{(1 - \rho)^2\nu^2}\left( \frac{\mathcal{L}(\mathbf{x}^{(0)}) - \mathcal{L}^*}{\gamma nT} + \left( \frac{\sqrt{1 - \beta_2}G'}{\epsilon^2} + \frac{\gamma L'}{2\epsilon^2} \right)\sigma^2 + G^2 \right) \\
= & 4(\sqrt{\beta_2}G' + \epsilon)\left( 1 + \frac{2L^2\gamma^2}{(1 - \rho)^2\nu^2} \right)\left( \frac{\mathcal{L}(\mathbf{x}^{(0)}) - \mathcal{L}^*}{\gamma nT} + \left( \frac{\sqrt{1 - \beta_2}G'}{\epsilon^2} + \frac{\gamma L'}{2\epsilon^2} \right)\sigma^2 \right) \\
& + \frac{8L^2\gamma^2(\sqrt{\beta_2}G' + \epsilon)}{(1 - \rho)^2\nu^2}G^2 \\
= & O\left( \frac{\mathcal{L}(\mathbf{x}^{(0)}) - \mathcal{L}^*}{\gamma T} \right) + O(\sigma^2) + O\left( \frac{\gamma^2}{(1 - \rho^2)\nu^2} \right)
\end{aligned}
\tag{129}
$$

which gives us the desired result.

■

### E.2 Convergence proof sketch for the variation of DAG-Adam algorithm

Similar as adapt-while-communicate versus adapt-then-communicate format, our proposed DAG-Adam algorithm also can have another style:

$$\mathbf{g}^{(t)} = \mathcal{W}\nabla\widehat{\mathbf{f}}(\mathbf{x}^{(t)}) + \frac{\nu}{\gamma}(I - \mathcal{W})\mathbf{x}^{(t)} \tag{130}$$

$$\mathbf{m}^{(t)} = \beta_1\mathbf{m}^{(t-1)} + (1 - \beta_1)\mathbf{g}^{(t)} \tag{131}$$

$$\mathbf{v}^{(t)} = \beta_2\mathbf{v}^{(t-1)} + (1 - \beta_2)\mathbf{g}^{(t)} \odot \mathbf{g}^{(t)} \tag{132}$$

$$\mathbf{x}^{(t+1)} = \mathbf{x}^{(t)} - \gamma(\mathbf{H}^{(t)} + \epsilon)^{-1}\mathbf{m}^{(t)} \tag{133}$$

Note there are two $\mathcal{W}$ in the augmented gradient step, but we only need one round of communication in practice. To see that, we can re-write into this form

$$\mathbf{g}^{(t)} = \frac{\nu}{\gamma}\mathbf{x}^{(t)} - \frac{\nu}{\gamma}\mathcal{W}\left(\mathbf{x}^{(t)} - \frac{\gamma}{\nu}\nabla\widehat{\mathbf{f}}(\mathbf{x}^{(t)})\right) \tag{134}$$

That is, instead of communicating the parameters directly, we communicate the parameters with one SGD step forward.

To establish the convergence analysis of this algorithm, we need to find the gradient (134) as a standard gradient of another augmented loss function. As long as as we find that loss function, the rest of proof will be similar as we did in Theorem 1. Fortunately, it has been stated in Yuan et al. (2021) paper for the momentum algorithms. Here, we give a modification for the Adam algorithm.

First, we need to further assume that $W$ is a positive definite matrix. It is easy to achieve that since if the original $W$ is not, we can use $\tilde{W} = (I + W)/2$ instead. Suppose $W$ has eigenvalue decomposition form $W = Q^T\Lambda Q$, where $Q$ is an orthogonal matrix. Then, we define $\mathcal{W}^\alpha = Q^T\Lambda^\alpha Q$, which is also positive definite matrix for any $\alpha > 0$. So we have $W = W^{\frac{1}{2}}W^{\frac{1}{2}}$. Now we introduce the algorithm into a transformed domain by multiplying $\mathcal{W}^{-\frac{1}{2}}$ on the both sides:

$$\mathcal{W}^{-\frac{1}{2}}\mathbf{g}^{(t)} = \mathcal{W}^{\frac{1}{2}}\nabla\widehat{\mathbf{f}}(\mathbf{x}^{(t)}) + \frac{\nu}{\gamma}(I - \mathcal{W})\mathcal{W}^{-\frac{1}{2}}\mathbf{x}^{(t)} \tag{135}$$

$$\mathcal{W}^{-\frac{1}{2}}\mathbf{m}^{(t)} = \mathcal{W}^{-\frac{1}{2}}\beta_1\mathbf{m}^{(t-1)} + (1 - \beta_1)\mathcal{W}^{-\frac{1}{2}}\mathbf{q}^{(t)} \tag{136}$$

$$\mathbf{v}^{(t+1)} = \beta_2\mathbf{v}^{(t)} + (1 - \beta_2)\mathbf{g}^{(t)} \odot \mathbf{g}^{(t)} \tag{137}$$

$$\mathcal{W}^{-\frac{1}{2}}\mathbf{x}^{(t+1)} = \mathcal{W}^{-\frac{1}{2}}\mathbf{x}^{(t)} - \gamma\mathcal{W}^{-\frac{1}{2}}(\mathbf{H}^{(t)} + \epsilon)^{-1}\mathcal{W}^{\frac{1}{2}}\mathcal{W}^{-\frac{1}{2}}\mathbf{m}^{(t)} \tag{138}$$

We further introduce

$$\mathbf{s}^{(t)} \triangleq \mathcal{W}^{-\frac{1}{2}}\mathbf{x}^{(t)}, \mathbf{q}^{(t)} \triangleq \mathcal{W}^{-\frac{1}{2}}\mathbf{g}^{(t)}, \mathbf{p}^{(t)} \triangleq \mathcal{W}^{-\frac{1}{2}}\mathbf{m}^{(t+1)}, \quad \mathbf{U}^{(t)} \triangleq \mathcal{W}^{-\frac{1}{2}}\mathrm{diag}(\mathbf{v}^{(t)})\mathcal{W}^{\frac{1}{2}} \tag{139}$$

The algorithm (130)-(133)is equivalently to

$$\mathbf{q}^{(t)} = \mathcal{W}^{\frac{1}{2}}\nabla_{(\cdot)}\widehat{\mathbf{f}}(\mathcal{W}^{\frac{1}{2}}\mathbf{s}^{(t)}) + \frac{\nu}{\gamma}(I - \mathcal{W})\mathbf{s}^{(t)} \tag{140}$$

$$\mathbf{p}^{(t)} = \beta_1\mathbf{p}^{(t-1)} + (1 - \beta_1)\mathbf{q}^{(t)} \tag{141}$$

$$\mathbf{U}^{(t)} = \beta_2\mathbf{U}^{(t-1)} + (1 - \beta_2)\mathcal{W}^{-\frac{1}{2}}\mathrm{diag}\left[(\mathcal{W}^{\frac{1}{2}}\mathbf{q}^{(t)})\right]^2\mathcal{W}^{\frac{1}{2}} \tag{142}$$

$$\mathbf{s}^{(t+1)} = \mathbf{s}^{(t)} - \gamma(\mathbf{U}^{(t)} + \epsilon)^{-1}\mathbf{p}^{(t)} \tag{143}$$

where the notation $\nabla_{(\cdot)}f(\cdot)$ means taking the gradient with respect to the whole argument of $f$ and note that $\mathcal{W}^{-\frac{1}{2}}(\mathbf{H}^{(t)} + \epsilon)^{-1}\mathcal{W}^{\frac{1}{2}} = \left(\mathcal{W}^{\frac{1}{2}}\mathbf{H}^{(t)}\mathcal{W}^{-\frac{1}{2}} + \epsilon\right)^{-1}$. Remember we can always retrieve $\mathbf{x}^{(t)}$ by $\mathbf{x}^{(t)} = \mathcal{W}^{\frac{1}{2}}\mathbf{s}^{(t)}$. Now, we are prepared for introduce this new augmented loss function

$$\mathcal{L}_W(\mathbf{s}) = \mathbf{f}(\mathcal{W}^{\frac{1}{2}}\mathbf{s}) + \frac{\nu}{2\gamma}\|\mathbf{s}\|_{I-W}^2 \tag{144}$$

It is not hard to verify that

$$\nabla_\mathbf{s}\mathcal{L}_W(\mathbf{s}) = \mathcal{W}^{\frac{1}{2}}\nabla_{(\cdot)}\mathbf{f}(\mathcal{W}^{\frac{1}{2}}\mathbf{s}^{(t)}) + \frac{\nu}{\gamma}(I - \mathcal{W})\mathbf{s}^{(t)} \tag{145}$$

This means that (140) - (143) can be viewed as Adam algorithm applied on this augmented loss function $\mathcal{L}_W(\mathbf{s})$. There is one difference that $\mathbf{U}^{(t)}$ is no longer diagonal matrix. However, recall that we mainly use it in weighted norm, and the useful inequality (25) can be extended for arbitrary positive matrix

$$a\|x\|_A^T \leq \|Ax\|^2 \leq b\|x\|_A^T \tag{146}$$

where $a$ and $b$ are the smallest and largest eigenvalue of matrix $A$. To see that, suppose $A = Q^T \Lambda Q$, we have

$$\|Ax\|^2 = \|Q^T \Lambda Q x\|^2 = \|\Lambda Q x\|^2 \leq b\|\Lambda^{1/2} Q x\|^2 = b\|x\|_A^2 \tag{147}$$

Therefore, we can establish the similar conclusion as Theorem 1.

## F  Heterogeneous data distribution generation

This subsection provides the detailed heterogeneous data generation for CIFAR-10 training dataset. To simulate the heterogeneous data, the disjoint non-i.i.d. training data over agents are generated through the Dirichlet distribution. For each data class, a categorical vector $\mathbf{q} = (q_1, \cdots, q_n)$, where $\sum_i q_i = 1, q_i \geq 0$, is used to illustrate the training data distribution in a network of size $n$. It is drawn from a symmetric Dirichlet distribution $\text{Dir}(\alpha)$, where $\alpha$ stands for the degree of the data heterogeneity (Lin et al., 2021; Yurochkin et al., 2019). The training data for a particular class tends to concentrate in a single node as $\alpha \to 0$, i.e. becoming more heterogenous while the homogeneous data is formed as $\alpha \to \infty$. After the sampling procedure, the data is reallocated to ensure the same amount of training data in each network node.

Fig. 9 visualizes the heterogeneous data distribution for the CIFAR-10 experiments shown in Table 1 for both network size (4 and 8). The radius of each circle stands for the proportion of the samples for the corresponding data class in the node. Clearly, as $\alpha \to 0$, the data for each class tends to be allocated in a single node. When $\alpha \to \infty$, all the data is distributed evenly.

Fig. 10 illustrates the training loss and validation accuracy curves of different algorithms when $\alpha = 10$. It is observed in the left plot that PAdam and DAG-Adam converges with much lower training losses and higher validation accuracy than the others due to their insensitivity to data heterogeneity.

## G  More experimental setting and BERT Experimental Results

**More Implementation details.** We also follow DDP's design to enable computation and communication overlap. Each server contains 8 V100 GPUs in our cluster and is treated as one node. The inter-node network fabrics are 25 Gbps TCP as default, which is a common distributed training platform setting. The exponential topologies (Assran et al. (2019)) are utilized as default.

**Training Loss.** Fig. 11 shows the iteration-wise training loss curves of aforementioned algorithms separately.

**The effect of using dynamic $\nu$.** Besides with fine-tuning a proper $\nu$ constant for DAG-Adam, we tried to use a dynamic $\nu$ scheduler to validate its performance. Compared to the common learning rate scheduler, we use dynamic $\nu$ schedulers that provide a relatively small $\nu$ value at first and grow to 1 at the end of training. Both linear schedulers and cosine schedulers were examined. Similar as learning rate scheduler, we also use a warm-up strategy, i.e., to fix a static $\nu$ for a certain iterations then use dynamic $\nu$ schedulers afterwards. We provide some preliminary results in Table 6. Unfortunately, it did not provide better performance compared with the result of using constant $\nu$. We leave it as future works to design a better strategy to select $\nu$ adaptively without introducing more hyper-parameters.

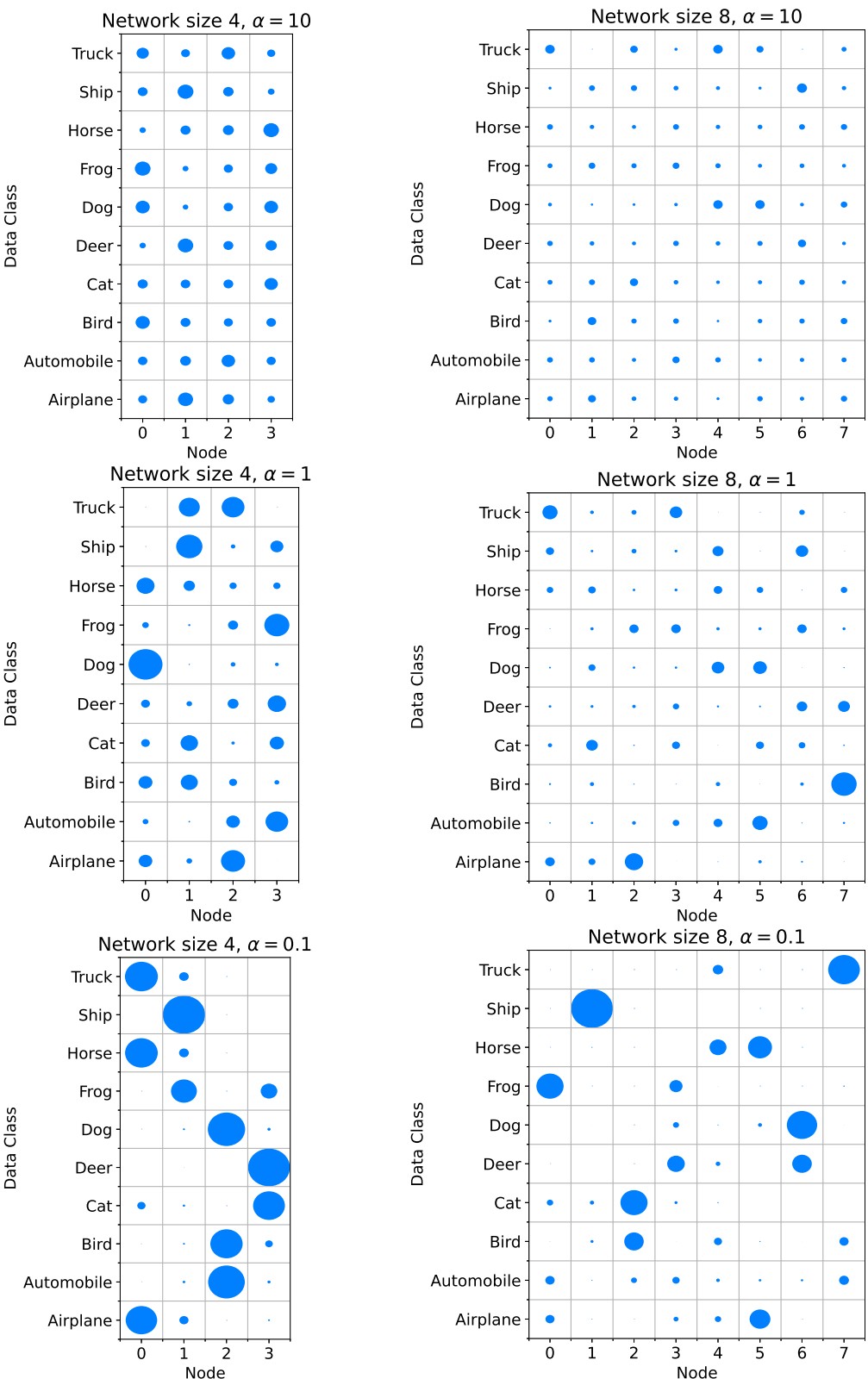

Figure 9: Heterogeneous data distribution visualization for CIFAR-10 dataset.

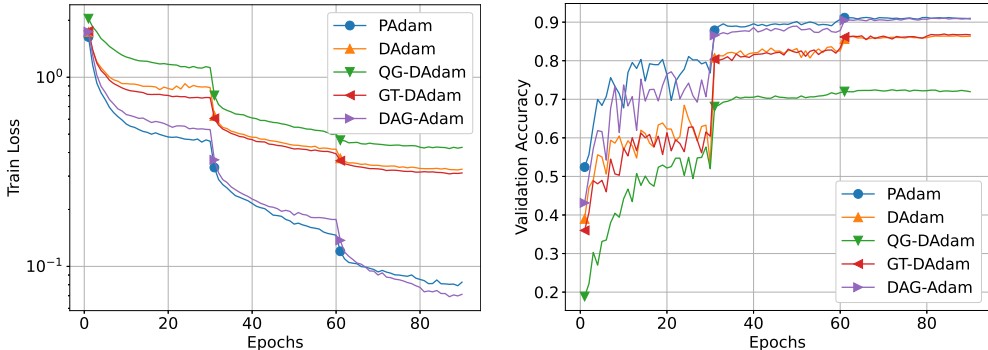

Figure 10: Training loss and validation accuracy evolution for CIFAR-10 dataset under a heterogeneous network of size 8 with $\alpha = 10$. Left plot: training loss. Right plot: validation accuracy.

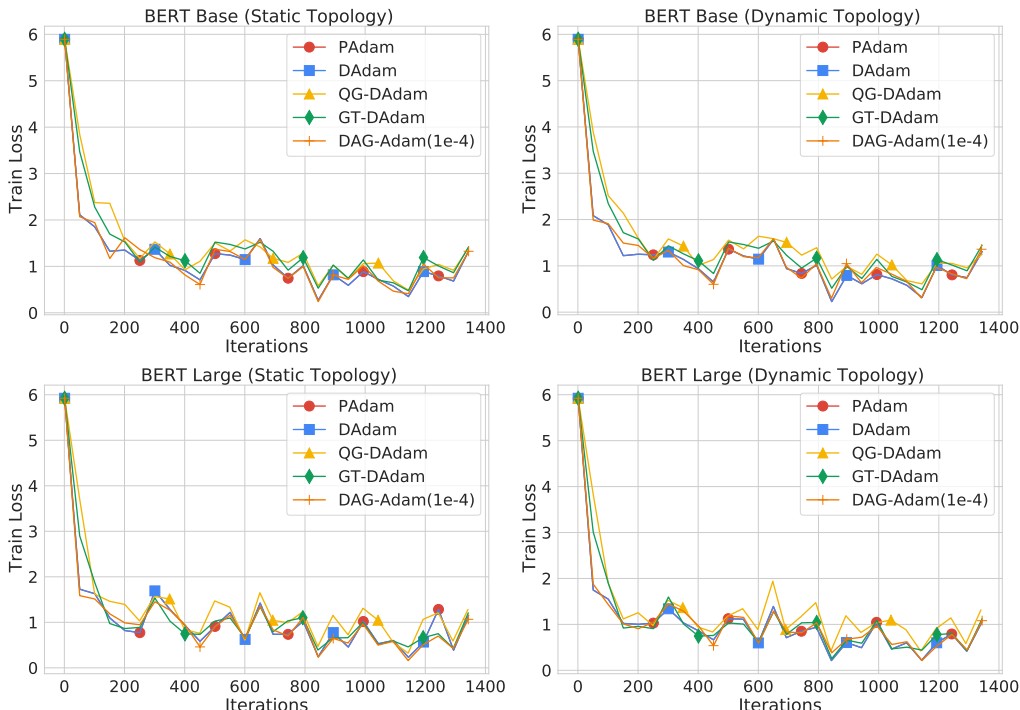

Figure 11: Convergence results on the BERT fine-tuning (SQuAD) task with different models and topologies.

Table 6: The effect of using dynamic $\nu$ (BERT base with static topology).

| SCHEDULER | WARM-UP RATIO | EXACT MATCH | F1 |
|---|---|---|---|
| LINEAR | 0 | 46.68 | 58.64 |
| COSINE | 0 | 65.28 | 75.99 |
| LINEAR | 0.2 | 76.26 | 84.69 |
| COSINE | 0.2 | 77.97 | 85.85 |
| LINEAR | 0.35 | 80.26 | 86.69 |
| COSINE | 0.35 | 80.97 | 87.85 |

