# OpenReview forum: "Communicate Then Adapt: An Effective Decentralized Adaptive Method for Deep Training"
_ICLR.cc/2022/Conference — ICLR 2022 Submitted_

### Official Review · Reviewer_XYfg · 2021-10-31

**Correctness:** 3
**Technical Novelty And Significance:** 2
**Empirical Novelty And Significance:** 2
**Recommendation:** 3
**Confidence:** 5

**Main Review:**

Pros:
1. The motivation is clear. It shows why existing methods do not converge to the stationary point.

2. The extensive experiments can support the superiority of this new method.

Cons:

1. The novelty is not significant. It is an extension of (Yuan et al., 2016) to the adaptive gradient.

2. This method can only converge to the neighborhood of the stationary point. However, existing methods can converge to the stationary point.

3. $\beta_2$ is lower bounded. It is not very reasonable. $\epsilon$ should be very small. Then, this lower bound is very large.
It conflicts with the common practice that $\beta_2=0.1$ or $\beta_2=0.01$. Thus, this convergence rate is problematic.

4. How is $\beta_1$? Is there any constraint for $\beta_1$?

5. Which value is used for $\beta_2$ in the experiment?

5. Which value is used for $\epsilon$ in the experiment? Typically, $\epsilon$ should be very small, e.g., $1e-7$. According to Theorem 1, $v$ should be smaller than $\epsilon$. But it looks $v$ is large in the experiment. Thus, I don't think the hyperparameters in Theorem 1 are reasonable.


**Summary Of The Paper:**

This paper developed a new decentralized adaptive gradient descent method to address the data heterogeneity problem. The motivation is clear and the experimental results show improvement over existing methods. However, the theoretical analysis is not  solid.

**Summary Of The Review:**

The problem studied is interesting. But the method is not novel and the theoretical analysis is not solid.

---

> ### Author Response · Authors · 2021-11-18
> **Response to Reviewer XYfg (Part III)**
>
> **[Q6. Unreasonable hyper-parameter]** Many thanks for the careful review!
>
> * Please note that we establish $\nu \le \epsilon$ to be one **sufficient condition** to guarantee the convergence of DAG-Adam. **It does not necessarily imply that a larger $\nu$ cannot guarantee DAG-Adam to converge**. In theoretical analysis (especially in adaptive gradient method which is notoriously difficult to analyze), it is quite common that the established convergence condition is conservative. The real practical value for some quantities can be set much larger than what the theory shows.  Below are some examples in literatures that are well-cited or from well-established groups. With these examples, we hope the reviewer can understand the gap between the theoretical analysis and the real implementation does exist in many literatures, and it is not an uncommon phenomenon.
>
>     * In (Zaheer et al., 2018), the learning rate is established as $\gamma \le \epsilon/(2L)$ to guarantee convergence. However, the learning rate in the real experiment has never been that small during all iterations.
>
>     * In (Reddi et al., 2020) has the same situation as in (Zaheer et al., 2018).
>
>     * In (Lin et.al., 2021), it is established that the momentum parameter $\beta$ has to stay within (0, 1/21) to guarantee linear speedup. However, the real value used for $\beta$ is 0.9 or 0.99 in real experiment. In addition, this paper clearly explains that "our
> theorem imposes some constraint on the momentum parameter. In practice, however, the method performs well even when this constraint is violated."
>
>
> * On the other hand, the contribution of our work, as we summarized in the introduction, lies on the identification of convergence issues in existing works, development of effective method that can overcome the issue, providing analysis to show the effectiveness of our algorithm, and conducting extensive convex and deep learning experiments to corroborate the superiority over existing methods. We did not claim the tightness in the current analysis. Furthermore, since our algorithm is the first to incorporate consensus-enforced terms into adaptive momentum buffer, it is new and very challenging. At this stage, It is very difficult for us to provide tighter bounds on $\nu$ and other quantities.
>
>
> * However, we do agree that the loose bound for $\nu$ is a limitation in our current analysis. We will add a remark after Theorem 1 to acknowledge such limitation, and explicitly clarify the gap between theory and the experimental implementation. Hope this can resolve your concern on such hyper-parameter.
>
> **[Summary].** In this rebuttal, we have attempted to address all your concerns as best as we can. In particular, we listed our contribution and novelty, clarified some of your misunderstanding on our  parameter setting in $\beta$, and explained why our analysis is solid. We also acknowledge several limitations pointed out by the reviewer and will have explicitly statement or remark on them in the revision. We will be very glad to address any further comments or questions.

---

> ### Author Response · Authors · 2021-11-18
> **Response to Reviewer XYfg (Part II)**
>
> Note: all references mentioned below can be found in our paper.
>
> **[Q2. Neighborhood convergence]** Thank you for raising this question. We are happy to clarify it.
> * It is true that the proposed method can only converge to the neighborhood. As we clarified in Remark 2, such a non-vanishing error $O(\sigma^2)$ in DAG-Adam is caused by utilizing the vanilla Adam algorithm as a building block, **not the decentralized structure (communicate-then-adapt) proposed in our work**. The worst-case inexact convergence of vanilla Adam is well-studied in literature. For example, (Reddi et.al., 2019) shows that vanilla Adam will not converge to the exact solution in the worst case, and it also constructs a convex example to illustrate it. Another work (Zaheer et.al., 2018) explicitly shows that vanilla Adam will suffer from a constant non-vanishing term $O(\sigma^2)$ as what we have in Theorem 1. In fact, having non-vanishing term in convergence bound is not uncommon in Adam-based algorithms, see theorem 2 in Chen et al. (2021a) and theorem 3.1 in Chen et al. (2018b). **No existing works can establish exact convergence of methods built upon vanilla Adam to our knowledge**.
>
>
> * Such non-vanishing $O(\sigma^2)$ term can be easily corrected with existing techniques. For example, we can utilize increasing batch-size to gradually reduce $\sigma^2$ to 0 as (Zaheer et.al., 2018) suggested. (Reddi et.al., 2019) proposed AMSGrad to correct the $O(\sigma^2)$ term. More techniques are discussed in section "related work on adaptive gradient method" in our paper. However, it is worth noting that our work focuses on removing the non-vanishing bias (i,e, the $B^2$ term shown in Proposition 1) caused by the decentralized update. It is **orthogonal** to existing exactly convergent adaptive methods such as AMSGrad. For example, DAG-Adam + increasing batch-size can achieve exact convergence without any non-vanishing term. However, DAdam + increasing batch-size cannot correct the bias $B^2$ introduced by its adapt-while-communicate structure, see Proposition 1.
>
>
> * Finally, the inexact convergence in Adam is based on the worst-case analysis. In practice, Adam is widely-utilized to train computer vision and natural language tasks and shows strong empirical performance. This is another motivation that we develop decentralized methods based on the vanilla Adam.
>
>
> **[Q3. Problematic convergence rate]** Thank you for raising this question. We are happy to clarify it. As you correctly commented, $\beta_2 \ge 1 - c \epsilon^2$ (constant $c$ can be referred to Theorem 1) and the lower bound is close to 1. However, **it does not conflict with the common practice in which $\beta_1 = 0.9$ and $\beta_2 = 0.999$** (not $\beta_2 = 0.01$ as mentioned by the reviewer). Such setting is used in two highly-cited papers (Kingma and Ba. 2014) and (Reddi et.al., 2019), and it is also adopted in [Pytorch official document](https://pytorch.org/docs/stable/generated/torch.optim.Adam.html#torch.optim.Adam). Since $\beta_2$ is chosen very close to $1$, it will not break the lower bound assumed in Theorem 1. Hope this clarification can resolve your confusion (we guess your impression that $\beta_2=0.01$ comes from the update $v^{(t)} = \beta_2 v^{(t-1)} + (1-\beta_2) g^{(t)} \odot g^{(t)}$, which is different from the update we utilized in DAG-Adam.).
>
>
> **[Q4: Constraint on $\beta_1$]** We set $\beta_1 = 0$ in Theorem 1 to simplify the convergence analysis. This is because the main challenge in the convergence analysis in adaptive methods is from the non-linear operation $g^{(t)}/\sqrt{v^{(t)}}$, not the momentum update $m^{(t)} = (1-\beta_1) m^{(t-1)} + \beta_1 g^{(t)}$. Assuming $\beta_1 = 0$ will not circumvent the difficulty handling the non-linear operation but can significantly simplify our analysis. In fact, many well-cited papers assume $\beta_1 = 0$ in analysis such as  (Zaheer et.al., 2018) and (Reddi et.al., 2020), and the techniques to deal with $\beta_1 \neq 0$ can be found in (Chen et.al., 2018b). Since the analysis of the paper focuses on handling the challenge brought by communicate-then-adapt structure, we respectfully believe such a simplification will not affect the major contribution of our analysis. However, we will explicitly assume $\beta_1 = 0$ in Theorem 1 to avoid any misunderstanding in the revision.
>
> **[Q5: $\beta_2$ in experiments]**  We set $\beta_2 = 0.999$ as recommended by (Kingma and Ba. 2014), (Reddi et.al., 2019) and [Pytorch official document](https://pytorch.org/docs/stable/generated/torch.optim.Adam.html#torch.optim.Adam).

---

> > ### Comment · Reviewer_XYfg · 2021-11-18
> > **$\beta_1$ and $\beta_2$**
> >
> > I suggest the authors check the original paper of Adam to see the difference between your $\beta_i$ and their $\beta_i$.
> >
> > In this paper, the coefficient for $m$ (or $v$) is $(1-\beta_i)$. On the contrary, the coefficient for in standard Adam is $\beta_i$.
> >
> > Thus, your argument on this hyperparameter does not hold.

---

> > > ### Author Response · Authors · 2021-11-18
> > > **Thanks for pointing out the typos in the algorithm listings.**
> > >
> > > We truly appreciated that the reviewer pointed out the different usages of $\beta$ and $1-\beta$. Now, we realized that we had misunderstood your comments on the value range of $\beta$.  We sincerely apologize for the equations listed in Algorithm 1, 2, and 3. These are indeed typos. We have fixed the order in the revision.  We have checked the main proof in the appendix. It is based on the same $\beta$ usage as the original Adam paper. As shown in the equations with the stacked vector forms (20)-(23), (96)-(101), and (130)-(133), all these equations are all in the standard $\beta$ usage and our proof is based on these stacked vector forms only. In the proof, a few inequalities are also based on the standard $\beta$ form.  For example, the term $\sqrt{\beta_2} H^{(t-1)}$ is introduced is because  $\sqrt{\beta_2} H^{(t-1)} \leq H^{(t)}$ holds when the $\beta$ multiplies over the historical average term instead of the latest one. Hence, our boundary on the hyperparameter and $beta_2$ still holds. Also, we have double checked that our experiment was executed based on the standard form of Adam algorithm. Again, we want to express our gratitude for the careful review and meaningful suggestions, and we will check the whole manuscript thoroughly one more time to make sure this typo doesn’t impact any other established theorems.

---

> ### Author Response · Authors · 2021-11-18
> **Response to Reviewer XYfg (Part I)**
>
> Many thanks for the valuable comments! We have attempted to address them as best as we can. We will be glad to clarify any further comments.
>
> **[Q1. Novelty]** While our algorithm is built upon (Yuan et.al., 2016) as you correctly commented, we would like to make several clarifications on the novelty.
>
> * **[Identifying issues in existing work]** The development methodology behind most existing decentralized momentum and adaptive methods is *standard single-node optimizer (such as standard SGD, momentum SGD, Adam) + gossip averaging* (which amounts to the adapt-then-communicate structure discussed in our paper). In such methodology, only the stochastic gradient is incorporated into the momentum update, and the gossip averaging that enforces consensus is decoupled from the single-node optimizer. While such methodology is easy to implement, it suffers from convergence issues as we clarified in Sec. 2 of our paper. We believe it is a non-trivial contribution to identify the convergence issue in such methodology.
>
>
> * **[Motivation behind DAG-Adam]** While the interpretation of decentralized SGD as standard SGD shown in (Yuan et.al., 2016) is a well-known result, very limited works utilize it as a tool, i.e., *incorporate both stochastic gradient and the consensus-enforced term $(I-W)x$ into the adaptive momentum update*,  to develop adaptive momentum methods. We believe it is because the drawback of the traditional methodology (standard optimizer + gossip averaging) is not well clarified before us. While DAG-Adam is built upon (Yuan et.al., 2016), we hope the reviewer can understand and value the reason and motivation why we develop the algorithm in this way. We clearly explained the new design philosophy (i.e., the adapt-then-communicate structure) can correct the convergence issue of the traditional methodology, which we believe is another solid contribution to the community.
>
>
> * **[Analysis]** Incorporating the consensus-enforced term $(I-W)x$ into the momentum $m^{(t)}$ and adaptive second-order moment $v^{(t)}$ update brings significant challenge to the analysis. With the non-linear operator $m^{(t)}/\sqrt{v^{(t)}}$ and the fact that both $m^{(t)}$ and $v^{(t)}$ involves $W$, the weight matrix in DAG-Adam will also go through a non-linear transformation and hence the effective weight matrix will become very complicated (see Appendix D in the revised paper for its formulation). Our analysis is the first that can overcome such difficulty and establish the convergence property of DAG-Adam.
>
>
> * **[Introduction of $\nu$]** DAG-Adam is not a trivial extension of (Yuan et.al., 2016) to the adaptive setting. An important contribution to boost the convergence performance of DAG-Adam is the **introduction of the hyper-parameter** $\nu$. Note that the vanilla augmented gradient in (Yuan et.al., 2016) is of the form of Eq. (13) in our paper. However, as we have explained in Appendix D in the revised paper, incorporating the consensus-enforced term $(I-W)x$ into the update of $m^{(t)}$ and $v^{(t)}$ (i.e., the non-linear transformation of $W$) will reduce the effectiveness to aggregate information in gossip averaging. To compensate such negative influence , we have to introduce $\nu$ to enable an effective  gossip averaging. In our response [Q2-3-1] to Reviewer 3, you can find a simple example in which how we leverage $\nu$ to achieve the best gossip averaging in DAG-Adam. We have added Appendix D in the revised paper to explain the influence of $\nu$ in detail.

---

### Official Review · Reviewer_RKoY · 2021-10-31

**Correctness:** 2
**Technical Novelty And Significance:** 2
**Empirical Novelty And Significance:** 2
**Recommendation:** 5
**Confidence:** 4

**Main Review:**

On one hand, I think the idea of DAG-Adam looks interesting, as it requires no additional computation -- just a reordering on the adapting and communication, the algorithm is guaranteed to converge to the exact solution on strongly convex problems. However, in terms of the correctness and insights, I have three main concerns:

- Both proposition 1 and 2 are shown in terms of the $O(\cdot)$, which does not really substantiate the claim "DAdam cannot converge to $x^*$" since 1) $a=O(\cdot)$ means $a\leq C\cdot$ for some constant $C$, but $a$ can be arbitrarily small; 2) With no lower bound, it's hard to argue a $O(\cdot)$ is tight, so it's possible that the extra term $G^2/\epsilon^2$ is incurred by a loose analysis. I think the only way to substantiate the claim is to derive a lower bound.

- The idea of DAG-Adam, different from the DSGD which directly follows from the update rule, requires an additional hyperparameter $\nu$. This essentially seems to reduce the mixing time of the original communication matrix $W$ -- which is straightforward to verify by Equation (13). This may be unfair in the comparison to other algorithms -- it is possible that DAdam can perform equally good if the mixing matrix there is also tuned by $\nu$. So I think more illustration is need there, especially in the experiments.

- The paper provides empirical studies on multiple tasks, including image classification and BERT finetuning. However, it's not clear to me how these results interact with the main theorem. As these are non-convex, and there is no $x^*$ anymore, it's unclear whether these improvement comes from the strategy of "Communicate Then Adapt", or just the fact that gradient tracker is used/mixing matrix is tuned. On a side note, I recommend the authors to include std for each table, as the results are pretty close.

**Summary Of The Paper:**

This paper proposes a new variant to decentralized Adam with a strategy called Communicate Then Adapt. The paper provides analysis and toy example to illustrate why a traditional Adam would fail to converge to the exact solution, and provide experiments on CV/NLP tasks to substantiate the theory.

**Summary Of The Review:**

Please refer to the main review. I'd be happy to increase my score if these concerns are properly addressed.

---

> ### Author Response · Authors · 2021-11-17
> **Response to Reviewer RKoY (Part II)**
>
> **[Q2-2. Influence of $\nu$ on DAdam]** Since DAdam is an immediate variant of DSGD, the influence of $\nu$ on DAdam will be similar to that on DSGD. After imposing the hyper-parameter $\nu$ to DAdam, we achieve the new recursion
>
> $
> x^{(t+1)} = ((1-\nu) I+\nu W) x^{(t)} - \gamma (H^{(t)} + \epsilon I )^{-1} m^{(t)}
> $
>
> and the update of the second term will not involve any communication. We add Fig. 8 in the revised paper to illustrate the influence of $\nu$ on DAdam. Similar to DSGD, it is observed that DAdam with $\nu=1$ (which is equivalent to the vanilla DAdam with no introduction of $\nu$) has the best convergence performance. **This empirical result shows that introducing $\nu$ to DAdam will not improve its convergence rate**.
>
> **[Q2-3. Influence of $\nu$ on DAG-Adam]** From equations (96) -- (100) in the revised paper, we can rewrite DAG-Adam as
>
> $
> x^{(t+1)}  =  x^{(t)}  - (1-\beta_1)\gamma (H^{(t)} + \epsilon I )^{-1} g^{(t)} + \beta_1(x^{(t)} - x^{(t-1)} ) + \beta_1 \Big(\gamma(H^{(t-1)} + \epsilon I )^{-1} - \gamma(H^{(t)} + \epsilon I )^{-1}\Big) m^{(t-1)}
> $
>
> In the following, we will first study a simple example that can shed lights on the influence of $\nu$ on DAG-Adam, and next we will discuss the more generalized scenario.
>
> **[Q2-3-1. An illustrating example]** We consider a trivial scenario in which $H^{(t)} = \alpha I$, which amounts to the case where all nodes maintain constant $v_i^{(t)}$ across $i$ and $t$ in the local buffer. As we have shown in the convex experiment in Fig. 5 in the revised paper, this trivial scenario becomes valid in the asymptotic stage of DAG-Adam in which $v_i^{(t)}$ converges to the same value $v$ for any $i \in [n]$. Substituting $H^{(t)} = \alpha I$ into the above recursion, we have
>
> $
> x^{(t+1)}  = \underbrace{\Big[ \big(1 - \frac{1-\beta_1}{\nu(\alpha + \epsilon)}\big) I + \frac{1-\beta_1}{\nu(\alpha + \epsilon)} W \Big]}_{:=\widetilde{W}} x^{(t)} - \frac{\gamma(1 - \beta_1)}{\alpha + \epsilon} \nabla f(x^{(t)}) + \beta_1(x^{(t)} - x^{(t-1)} )
> $
>
> From the expression of the effective weight matrix $\widetilde{W}$, it is observed that the optimal value for $\nu$ to minimize the mixing time of the weight matrix is $\frac{\alpha + \epsilon}{1-\beta_1}$, **which is not $1$** as in the vanilla DSGD algorithm. In addition, **it is worth noting that the effective weight matrix $\widetilde{W}$ in DAG-Adam cannot outperform $W$ in DAdam in terms of mixing time for any choice of $\nu$**.
>
> This trivial example illustrates the necessity to have the hyper-parameter $\nu$ in DAG-Adam. The introduction of $\nu$ is to compensate the negative influence of the communicate-then-adapt structure in DAG-Adam which incorporates both the gradient and the consensus constraint into the adaptive momentum buffer. **The weight matrix $\widetilde{W}$ can be quite inefficient to aggregate information if no $\nu$ exists to speed up partial  averaging**.
>
> **[Q2-3-2. General scenario].** In the general scenario in which $H^{(t)}$ is not constant across iterations, it is established in  Eq. (103) and (104) in the revised paper that the effective weight matrix of DAG-Adam is given by
>
> $
> \widetilde{W} = (I-\nu (1-\beta_1)(H^{(t)} + \epsilon I)^{-1})I +\nu (1-\beta_1)(H^{(t)} + \epsilon I)^{-1} W
> $
>
> From the above expression, it is difficult to determine the eigenvalue of $\widetilde{W}$, not to mention the optimal value for $\nu$. However, it is obvious that $\nu=1$ is not necessarily optimal for DAG-Adam. This motivates us to tune $\nu$ in  experiments to boost the convergence performance of DAG-Adam. In fact, the results in Table 1 (with CIFAR-10 dataset), Table 3 (with ImageNet dataset), and Table 4 (with SQuAD dataset) corroborate that $\nu=1$ is not the optimal choice for DAG-Adam. How to find the optimal $\nu$ automatically is the main direction of our future work.
>
> **[Q3-1. No $x^\star$ in non-convex scenario]**  Many thanks for pointing this typo out. We have changed $\mathcal{L}(\mathbf{x}^\star)$ to $\mathcal{L}^\star$, which is defined as $\mathcal{L}^\star = \arg\min_{x}\{\mathcal{L}(x)\}$. Please check Theorem 1 in the revised paper.
>
> **[Q3-2. Source of the improvements]** As we have explained in Q2, introducing $\nu$ to DAdam will not improve its convergence performance, and the mixing time of DAG-Adam cannot outperform that of DAdam even if the optimal $\nu$ is picked up (which is unpractical in real implementations). For these reasons, the improvement of DAG-Adam comes from the communicate-then-adapt structure which incorporates consensus constraints into the adaptive momentum buffer so that the solution deviation issue suffered by DAdam and QG-DAdam can be corrected. The newly-added Figure 8 illustrates how DAdam performs with different $\nu$, which excludes the possibility that DAdam can be further improved by tuning $nu$.
>
> **[Q3-3. Include std to the table]** We added std error in Table 1 and 2. Will add std to other tables in the revision

---

> ### Author Response · Authors · 2021-11-17
> **Response to Reviewer RKoY (Part I)**
>
> Thanks for the valuable questions! We have attempted to address them as best as we can. **We will be glad to clarify any further comments**.
>
> **Q1 [Lower Bound]** Many thanks for this useful question! We agree that a lower bound can better substantiate the claim that "DAdam cannot converge to $x^\star$". To address your concern, we have refined the upper bound in Proposition 1 and found a lower bound for $\sum_{i=1}^n \|x_i^{(t)} - x^\star\|^2$ when $t\to \infty$. For the reviewer’s convenience, we repeat the revised Proposition 1 as follows (the reviewer can also check the proposition in the revised paper):
>
>
> **Proposition 1**. *Under Assumptions  A.1,  A.3  and  A.4,  if  each $f_i(x)$ is  further  assumed  to  be $μ$-strongly-convex,  and  the  full-batch  gradient $\nabla f_i(x)$ is  accessed  per  iteration,  then  DAdam  (Algorithm 2) cannot converge to $x^\star$, i.e., the global solution to problem (1). In particular, the distance between DAdam limiting point and $x^\star$ can be characterized as follows (Proof is in Appendix B.3):*
>
> $
> \lim_{t\to \infty} \sum_{i=1}^n \|x_i^{(t)} - x^\star\|^2 = O\Big( \frac{\gamma^2 G^2}{(1-\rho)^2\epsilon^2} + B^2\Big)
> $
>
> *where $B := \|\frac{1}{n}\sum_{i=1}^n (H_i + \epsilon I)^{-1} \nabla f_i(x_i^\infty) - \frac{1}{n}\sum_{i=1}^n \nabla f_i(x_i^\infty)\| \neq 0$ is a constant bias term. **Furthermore, the $B^2$ term in the above bound is necessary and cannot be removed by an improved analysis; there exists strongly convex and smooth functions** $f_i(x)$ such that*
> $
> \lim_{t\to \infty} \sum_{i=1}^n \|x_i^{(t)} - x^\star\|^2 \ge B^2
> $
>
>
> With the newly-established lower bound in Proposition 1, it is observed that DAdam cannot converge to the optimal solution $x^\star$ even if a decaying learning rate $\gamma \to 0$ is utilized. We also revised Proposition 2 for QG-DAdam accordingly, please check the details of Proposition 2 in the revised paper.
>
> **[Q2. Influence of hyper-parameter $\nu$]** Thank you for this excellent question. We have added a new section in Appendix D to discuss the influence of $\nu$ on DSGD, DAdam, and DAG-Adam. Before going to the details, we list the summary here:
>
> * Introducing $\nu$ to DSGD or DAdam is not necessary. It will **not** reduce the mixing time of the weight matrix in these methods. In the following discussions, we will show that DSGD or DAdam with $\nu = 1$, i.e., with no hyper-parameter $\nu$ in the algorithm, will achieve best convergence performance theoretically or empirically. **In fact, any value $\nu \in [0, 1)$ will increase the mixing time of DSGD or DAdam and hence hurt the convergence**.
>
>
> * **Introducing $\nu$ to DAG-Adam is necessary**. DAG-Adam utilizes a communicate-then-adapt structure, which incorporates consensus constraints (i.e., $(I-W)x$), which involves the weight matrix $W$, into the adaptive momentum buffer $m^{(t)}$ and $v^{(t)}$. We will show such a communicate-then-adapt structure will hurt the spectral gap of $W$ and hence increase its mixing time. We have to choose an appropriate $\nu \in (0, 1)$ to compensate the inefficiency in gossip average in DAG-Adam.
>
>
> * **The improvement in DAG-Adam comes from its communicate-then-adapt structure** which was particularly developed to resolve the solution deviation issue suffered by DAdam and QG-DAdam. **It is not from the tuning of parameter $\nu$**. In fact, we will show the mixing time of the effective weight matrix in DAG-Adam cannot be smaller than that of DSGD or DAdam even if the optimal $\nu$ is chosen.
>
> Now we discuss the details. More detailed discussion can be found in Appendix D in the revised paper.
>
> **[Q2-1. Influence of $\nu$ on DSGD].** Recall equation (13), we can introduce parameter $\nu$ to DSGD as follows
>
> $
> x^{(t+1)} = x^{(t)}- \gamma \Big( \nabla f (x^{(t)}) + \frac{\nu}{\gamma} (I-W)x^{(t)} \Big) = \underbrace{\Big((1-\nu) I + \nu W\Big)}_{:=\widetilde{W}}x^{(t)} - \gamma \nabla f (x^{(t)})
> $
>
> in which we let $\widetilde{W}$ denote the **effective** weight matrix. Apparently, the $\nu$ value will influence the mixing time of $\widetilde{W}$. The eigenvalue of $\widetilde{W}$ is given by
>
> $
> {\rm eig}_k(\widetilde{W}) = (1-\nu)\cdot 1 + \nu \cdot {\rm eig}_k(W) \quad \Longleftrightarrow \quad 1 - {\rm eig}_k(\widetilde{W}) = \nu(1-{\rm eig}_k(W))
> $
>
> The above relation implies all eigenvalues of $\widetilde{W}$ are more closer to value 1 for any $\nu \in [0, 1)$. Since a larger spectral gap, i.e. $1-\lambda_2(\widetilde{W})$, will reduce the mixing time of gossip average and hence speed up the DSGD convergence, **the optimal choice for DSGD is $\nu=1$; other $\nu$ values will not improve the convergence of the vanilla DSGD**.
>
> We validate the above discussion with Fig. 7 in the revised paper. It illustrates the influence of $\nu$ on DSGD. It is observed that $\nu=1$ (which amounts to no introduction of $\nu$) enables DSGD to converge fastest and achieve the best validation accuracy, which is consistent with the above conclusions.

---

### Official Review · Reviewer_i1gD · 2021-11-02

**Correctness:** 4
**Technical Novelty And Significance:** 3
**Empirical Novelty And Significance:** 3
**Recommendation:** 8
**Confidence:** 3

**Main Review:**

In this paper authors propose to change the order of adaptation and communication in contrast with D-Adam. More precisely, First, in Section $2$, authors consider heterogeneous data as an input of D-Adam algorithm. They show that algorithm can diverge, moreover, under some additional assumptions it is proven to diverge. Another version of distributed Adam QG-DAdam uses another strategy - adapt-then-communicate. This algorithm also cannot converge to the optimum with heterogeneous input. Finally, they discussed the loss of efficiency of recent GT-DAdam in the same setup.
These arguments motivates authors to design the algorithm with communicate-then-adapt. This algorithm is based on the augmented gradient update that allows to interpret it as standard SGD algorithm with the same proof techniques.

In Theorem $1$, authors present the theoretical result for their algorithm that converges: with the constant learning rate the augmented gradient converges to 0; with $\gamma = 1/\sqrt{T}$ the global average of iterates converges to the stationary point.

Finally, in the experiments authors compare their algorithm with different modifications of DAdam (that diverge) and Distributed SGD and shows that in practice their algorithm also makes sence.


**Summary Of The Paper:**

In this paper, authors propose the novel modification of Distributed Adam Algorithm that allows heterogeneous data.

**Summary Of The Review:**

I want to thank authors for their work. The heterogeneous input is very important and I think that the adaptation of Adaptive Distributed Adam to it is an important step.
The only thing I am curios is the following: how do authors propose to calculate the global average and how long will it take in the end of the algorithm to calculate it?
Furthermore, I think that this result can be extended to the case when instead of weighted update with neighbors we select every neighbor with some probability and communicate only among them. This can save a lot of communication cost and make an algorithm even faster.

---

> ### Author Response · Authors · 2021-11-17
> **Response to Reviewer i1gD**
>
> Many thanks for the acceptance decision!
>
> **[Q1. Global average $\bar{x}$].** Thank you for raising this useful question. The global average $\bar{x}$ will be calculated by the Ring-Allreduce operation [R1, R2]. Ring-Allreduce is widely employed in PyTorch and Horovod to compute global average. Its communication time is typically 2 or 3 times as long as the neighborhood partial averaging used in this work. In fact, the communication time comparison between global averaging and partial averaging was reported in existing literature such as [Table 17, R3]. Hope these explanation and references can resolve your confusion.
>
> [R1] P. Patarasuk and X. Yuan, "Bandwidth Optimal All-reduce Algorithms for Clusters of Workstations", 2009
>
> [R2] A. Gibiansky, "Bringing HPC Techniques to Deep Learning"
>
> [R3] Y. Chen, et.al., "Accelerating Gossip SGD with Periodic Global Averaging", 2021
>
> **[Q2. Random neighbor selection]** Thank you for this insightful comment! We agree that a probabilistic neighborhood selection will further save more communications. Your suggested probabilistic neighborhood selection model, in our opinion, is equivalent to the utilization of the random and time-varying weight matrix $W^{(t)}$, which represents the random and time-varying connectedness between each pair of neighbors. Decentralized methods with time-varying weight matrix have been extensively studied in existing literatures, see, e.g., [R4]
>
> [R4] A. Koloskova, et. al. "A Unified Theory of Decentralized SGD with Changing Topology and Local Updates"
>
> As a result, we believe your suggestion is both novel and doable. We will leave it as a future work.
>
> Thank you again for the positive comments. We will be glad to clarify any further questions or comments.

---

### Official Review · Reviewer_1RX3 · 2021-11-03

**Correctness:** 2
**Technical Novelty And Significance:** 3
**Empirical Novelty And Significance:** 2
**Recommendation:** 5
**Confidence:** 4

**Main Review:**

Strengths:
I found this paper very interesting, and have been long awaiting progress on the topic of adaptive decentralized optimization for deep learning. Traditional gossip-based methods performing consensus on the parameters cannot be naively applied to adaptive gradient methods, such as Adam, because of the non-linearity in the gradient update (e.g., dividing by the square-root of the second moment buffer), which breaks traditional proofs and significantly degrades performance in practice compared to centralized learning (e.g., see [a]). This paper is very important and contributes to an emerging literature. In particular, to realize the benefits of decentralized (gossip-based) optimization for deep learning, one must develop strategies suited for adaptive gradient methods, especially since many popular deep learning tasks seem to only work with adaptive methods, such as language models and the new swath of image-based methods relying on Vision Transformer architectures.

This paper also shows a very good familiarity with related work. Despite my current rating, I strongly support this paper, and thus will focus on the main issues I observed below so that the authors can address them, and hopefully find their paper accepted at the end of this process.

Weaknesses:
- Proposition 1 is somewhat meaningless… what is relevant here is a lower bound on the squared norm of the sub-optimality of the iterate $x^t_i$ as $t$ goes to infinity. Similarly for Proposition 2.
- Can you clarify the following: Theorem 1 assumes that not just the gradient is bounded, but that the gradient with an additional consensus penalty is bounded?
- In Theorem 1, the language indicates that $x^\star$ is the optimal solution, but note that in the non-convex setting there does not necessarily exist a unique global optimizer.
- Would be interested in seeing the statistical significance of results (across a few seeds) in Table 1 on CIFAR10.
- Why does heterogeneity of data distributions have such a large impact on the final validation performance in Table 1? I would expect the heterogeneity to affect the convergence rate, but not the final validation performance.
- What architecture is used for the Image Classification experiments?
- I don’t find enough evidence to support the fact that adapt-the-combine is the issue preventing convergence. It is quite bizarre to me that there should be a difference in adapt-then-combine compared to combine-then-adapt, since the two are often trivial re-parameterizations for gossip-based methods. For example, consider your case of the adapt-then-combine structure (section 2.2) equation 8. Let z^{t+1}_j = x^{t}_j - \gamma \frac{m^t_j}{\sqrt{v^t_j} + \epsilon} and let x^{t+1}_i = \sum_{j \in N_i} w_{ij} z^{t+1}_j, then the $z$ variable has the same fixed point iteration as equation (16) in your combine-then-adapt setting. To me it seems that the reason your proposed method seems to resolve the obvious issue of adaptive methods with linear consensus updates is that the consensus update is incorporated into the adaptive momentum buffer (i.e., tracking not just moving average of gradients, but also the consensus constraint), as opposed to swapping the order of adapt-then-combine vs combine-then-adapt. On this note, I would urge you to also take a look at the SlowMo work of [b] which incorporates a slow-moving consensus step into the momentum update of decentralized gossip-based methods.
- There might be a problem with Figure 1 since none of the methods convergence to the optimal solution in this strongly-convex case. For logistic regression in particular there isn’t an analytic solution for $x^\star$ so I’m curious how you’re computing it? Moreover, depending on the conditioning, there could be several solutions in the argmax set. I would be interested in seeing the logistic loss plotted instead, and would be particularly interested in seeing the MSE of the variable z (which I mentioned in the bullet above) plotted for the adapt-the-combine methods.
- With respect to the fixed-point iteration on page 4: the condition $f_i(x) \neq f_j(x)$ does not necessarily guarantee $H_i \neq H_j$ unless you make additional assumptions about how f_i and f_j differ. E.g., consider an extreme case where the distribution differences between $D_i$ and $D_j $amount to a constant shift in the expected loss.

[a] Assran et al., Advances in asynchronous parallel and distributed optimization, Proceedings of the IEEE, 2020.

[b] Wang et al., SlowMo: Improving communication-efficient distributed SGD with slow momentum, ICLR, 2019.

**Summary Of The Paper:**

This paper proposes a decentralized adaptive method for distributed deep learning, termed DAG-Adam. Convergence results are provided for smooth non-convex objectives under a bounded gradient assumption. Numerical experiments are conducted on Image Classification (CIFAR10, ImageNet-1k) and Language Modelling (fine-tuning pre-trained BERT models on SQuAD).

**Summary Of The Review:**

In short, I find the paper very well motivated, the problem is very important, authors show a familiarity with related work, and the method is sufficiently novel.

However, I do not find the claims in the paper to be well supported. Specifically, I don’t find enough evidence to support the fact that adapt-then-combine is the issue preventing convergence of previous methods, and that the proposed method resolves this issue by using a combine-then-adapt strategy, especially since:
- with a simple re-parameterization, the proposed method can actually be written as an adapt-then-combine strategy
- propositions 1 and 2 showing issues with adapt-then-combine strategies provide upper bounds, but these are completely uninformative since the relevant quantities should provide a lower bound on the sub-optimality of the iterates with these strategies
It seems to me that the proposed method actually works because of the fact that consensus steps are incorporated into the first/second-order momentum buffers, not because the order of adapt/combine is flipped.

---

> ### Author Response · Authors · 2021-11-15
> **Response to Reviewer 1RX3 (Part III)**
>
> **Q8 [Problems with Fig. 1].**  Many thanks for raising this question.
>
> * For DAdam and QG-DAdam, it is established in Proposition 1 and 2 that the distance between the iterate and the solution $\frac{1}{n}\sum_{i=1}^n \|x_i^{(t)} - x^\star\|^2$ is upper bounded by $O(\gamma^2 + B^2)$. For GT-DAdam and DAG-Adam, it is established in Proposition 3 and 4 that the distance is on the order of $O(\gamma^2)$. For DSGD, it is established in (Chen and Sayed, 2012) that the distance is also on the order of $O(\gamma^2)$. When we utilize a constant learning rate $\gamma > 0$, it is observed from these bounds that none of these algorithms can converge to the optimal solution but to an error $O(\gamma^2)$. Note that even the standard SGD (with no decentralization) cannot converge to the optimal solution; it converges on the order of $O(\gamma^2)$.
>
> * To achieve the optimal solution $x^\star$, we utilize the full-batch standard gradient descent to solve the logistic regression for sufficient iterations and then evaluate the magnitude of the full-batch gradient at the convergent iterate $x^{(t)}$. If the full-batch gradient is very close to $0$ (i.e., within $10^{-8}$), we conclude such $x^{(t)}$ is an optimal solution (because it satisfies the optimality condition).
>
> * To address your concern on multiple solutions on the argmin set, we depict the convergence performance of all algorithms with respect to the **loss**, i.e., $\frac{1}{n}\sum_{i=1}^n\mathbb{E}[f(x_i^{(t)}) - f(x^\star)]$, see Fig. 6 in the appendix of the revised paper. We also depict the reparameterized DAdam you proposed in Q7 (which is named as DAdam-ReParam). It is observed that DAdam-ReParam converges closely to DAdam, and it does not have the same fixed point as DAG-Adam.
>
> * In the right plot of Fig. 6 in the revised paper, we depict the convergence of all algorithms using **stair-wise decaying learning rate**. It is observed both DAG-Adam and GT-DAdam indeed converged to the optimal solution under such learning rate settings as our propositions stated while DAdam and QG-DAdam cannot. DSGD didn't converge to the optimal solution is because such learning rate settings for Adam-type algorithms decay too fast for SGD one. All the observations in Fig. 1 and 6 are consistent with our derived bounds in Proposition 1 - 4.
>
>
> **Q9 [$H_i \neq H_j$].** Agree. Recall that $H_i:=\mathrm{diag}(\sqrt{\nabla f_i(x_i^\infty) \odot \nabla f_i(x_i^\infty)}\,)$, it we assume $\nabla f_i(x) \neq \nabla f_j(x)$ for any $x$ in the data heterogeneity scenario, it follows that $H_i \neq H_j$.
>
>
>
> **Summary.** We thank the reviewer again for his careful and insightful feedback. We have provided a detailed response for each question. In particular, we established a lower bound in Proposition 1 and 2, and had a detailed discussion on the communicate-then-adapt structure. We are happy to have follow-up discussions with you if there is any further question we can clarify.

---

> ### Author Response · Authors · 2021-11-15
> **Response to Reviewer 1RX3 (Part II)**
>
> **Q7 [Communicate-then-adapt structure].** We are very grateful for your insightful comments. We have tried our best to understand your arguments and clarify potential confusion. Please correct us if we misunderstood your idea.
>
>
> * First, we believe simply re-parameterizing recursion (8) into a communicate-then-adapt shape cannot converge to the same fixed point as our DAG-Adam. As you suggested, we let $x_i^{(t)} = \sum_{j=1}^n w_{ij} z_j^{(t)}$ and $z_i^{(t+1)} = x_i^{(t)} - \gamma \frac{m_i^{(t)}}{\sqrt{v_i^{(t)}} + \epsilon}$ where both $m_i^{(t)}$ and $v_i^{(t)}$ are calculated with respect to $x_i^{(t)}$. This new algorithm has a communicate-then-adapt structure. Now we examine its fixed point. Note that $x_i^\infty = \sum_{j=1}^n w_{ij} z_j^\infty$, $m_i^{(t)} = \nabla f_i(x_i^\infty)$, and $H_i = \mathrm{diag}\{\sqrt{\nabla f_i(x_i^\infty) \odot \nabla f_i(x_i^\infty)\}}$ for any $i\in [n]$. With these notations, the fixed point recursion of the newly-proposed communication-then-adapt algorithm is given by $z_i^\infty = \sum_{j=1}^n w_{ij} z_j^\infty - \gamma (H_i + \epsilon I)^{-1} \nabla f_i(x_i^\infty)$ for any $i\in [n]$ in which $x_i^\infty = \sum_{j=1}^n w_{ij} z_j^\infty$. If we assume $z_1^\infty=\cdots=z_n^\infty = z^\infty$, then it holds that $x_i^\infty = z^\infty$ and the fixed point $z^\infty$ needs to satisfy $\frac{1}{n}\sum_{i=1}^n (H_i + \epsilon I)^{-1} \nabla f_i(z^\infty) = 0$. Apparently, such a fixed-point $z^\infty$ is different from the fixed point of equation (16) which satisfies $\frac{1}{n}\sum_{i=1}^n \nabla f_i(x^\infty) = 0$. **In summary, the newly-proposed communicate-then-adapt algorithm cannot converge to the desired stationary solution when data heterogeneity holds**. We illustrate the convergence of reparameterized DAdam in Fig. 6. It is observed that reparameterized DAdam converges closely to DAdam; it converges to a less accurate solution than DAG-Adam when the same constant learning rate is utilized. Please correct us if we misunderstood your algorithms with re-parameterization. We are happy to clarify any further questions or comments.
>
>
> * Second, we use the communicate-then-adapt structure in the paper to distinguish DAG-Adam from existing approaches such as DAdam and QG-DAdam based on adapt-then/while-communicate structures, but it should not be the root reason to resolve the convergence issue suffered by DAdam as you correctly commented. For example, the above proposed reparamterized DAdam cannot converge to the desired solution even if it has a communicate-then-adapt structure. Consistent with your insights, the real technique in DAG-Adam that resolves the convergence issue is to **incorporate consensus updates into the adaptive momentum buffer**. While we did not clarify it explicitly, this technique is indeed a contribution made by DAG-Adam. We will add a remark to clarify the structure and the root reason why DAG-Adam works in the future revision.
>
>
> * Third, many thanks for bringing SlowMo to our attention. We will definitely cite it in the paper and carefully clarify the relation to it. Our understanding on SlowMo, and its relation to DAG-Adam, is as follows:
>
>     * SlowMo is built upon a base optimizer. After multiple rounds updated by the base optimizer, SlowMo will conduct a global synchronization and impose additional slow momentum to boost the generalization performance of the base optimizer. This base optimizer can be decentralized SGD, decentralized momentum SGD, DAdam, or DAG-Adam proposed in our paper. In this sense, SlowMo is **orthogonal** to DAG-Adam. Integration of DAG-Adam to SlowMo may lead to a better generalization performance, which will be left as our future work.
>
>     * DAG-Adam has just **one buffer** for momentum. While the consensus update (i.e. $\frac{\nu}{\gamma}(I-W)x$) is incorporated into  momentum, it does not require a new buffer for storage. Instead, the combo of gradient and the consensus update (i.e., $\nabla f(x) + \frac{\nu}{\gamma}(I-W)x$, which is of the same size as $\frac{\nu}{\gamma}(I-W)x$) will be stored in one buffer. On the other hand, the SlowMo strategy will impose **an extra momentum buffer** to the base optimizer, which is independent of the momentum buffer owned by the base optimizer (e.g., DAG-Adam). For this reason, DAG-Adam cannot be covered by SlowMo; they are orthogonal to each other。
>
>     *  Finally, we would like to clarify that SlowMo itself or SlowMo + DAdam/QG-DAdam cannot resolve the convergence deviation issue suffered by DAdam or QG-DAdam. It is because the SlowMo strategy will not affect the fixed point achieved by the base optimizer as established in (Wang et.al., 2020).

---

> ### Author Response · Authors · 2021-11-15
> **Response to Reviewer 1RX3 (Part I)**
>
> Thanks for the valuable comments! We have attempted to address them as best as we can. The reference [a] you mentioned has a consistent result with our work: gossip-based methods cannot be naively applied to adaptive gradient methods. We will cite and discuss [a] carefully in the future revision. We will be glad to clarify any further comments.
>
> **Q1 [Lower bound].** To address the reviewer’s concern, we have refined the upper bound in Proposition 1 and established a lower bound for $\sum_{i=1}^n \|x_i^{(t)} - x^\star\|^2$ when $t\to \infty$. For the reviewer’s convenience, we repeat the revised Proposition 1 as follows (the reviewer can also check the proposition in the revised paper):
>
> **Proposition 1**. *Under Assumptions  A.1,  A.3  and  A.4,  if  each $f_i(x)$ is  further  assumed  to  be $μ$-strongly-convex,  and  the  full-batch  gradient $\nabla f_i(x)$ is  accessed  per  iteration,  then  DAdam  (Algorithm 2) cannot converge to $x^\star$, i.e., the global solution to problem (1). In particular, the distance between DAdam limiting point and $x^\star$ can be characterized as follows (Proof is in Appendix B.3):*
>
> $
> \lim_{t\to \infty} \sum_{i=1}^n \|x_i^{(t)} - x^\star\|^2 = O\Big( \frac{\gamma^2 G^2}{(1-\rho)^2\epsilon^2} + B^2\Big)
> $
>
>
> *where $B := \|\frac{1}{n}\sum_{i=1}^n (H_i + \epsilon I)^{-1} \nabla f_i(x_i^\infty) - \frac{1}{n}\sum_{i=1}^n \nabla f_i(x_i^\infty)\| \neq 0$ is a constant bias term. **Furthermore, the $B^2$ term in the above bound is necessary and cannot be removed by an improved analysis; there exists strongly convex and smooth functions $f_i(x)$ such that** $\lim_{t\to \infty} \sum_{i=1}^n \|x_i^{(t)} - x^\star\|^2 \ge B^2$*.
>
> With the newly-established lower bound in Proposition 1, it is observed that DAdam cannot converge to the optimal solution $x^\star$ even if a decaying learning rate $\gamma \to 0$ is utilized. We also revised Proposition 2 for QG-DAdam accordingly, please check the details of Proposition 2 in the revised paper.
>
> **Q2 [Gradient Bounds].**  Yes, both the original gradient and the augmented gradient need to be bounded.
>
> **Q3 [$x^\star$ in Theorem 1].** Many thanks for pointing the typo out. We have changed $\mathcal{L}(\mathbf{x}^\star)$ to $\mathcal{L}^\star$, which is defined as $\mathcal{L}^\star = \arg\min_{x}\{\mathcal{L}(x)\}$. Please check Theorem 1 in the revised paper.
>
> **Q4 [Statistical significance].** Please check Table 1 in the revised paper.
>
> **Q5 [Data heterogeneity influence validation accuracy].** Many thanks for raising this useful question.
> * As you correctly commented, the data heterogeneity will slow down the convergence rate of decentralized algorithms. In the revised paper, we have added a new figure illustrating how different Adam-based algorithms converge with the CIFAR-10 training dataset, see Fig. 10 in the appendix. It is observed in the left plot that both PAdam and DAG-Adam converge faster than the others due to their insensitivity to data heterogeneity; they achieve much smaller training losses after the same amount of training epochs. This experiment corroborates your comments.
> * On the other hand, slow convergence rate does have a negative effect on the final validation (or test) performance. Note that all listed results in Table 1 are achieved with the **same resource budget**. In other words, they are obtained by utilizing the same hardwares, and updating models with the same amount of epochs (or iterations). If some algorithm converges slowly due to the influence of data heterogeneity, **the model may have not been sufficiently trained after the predefined number of training epochs**. As a result, the algorithm cannot converge to a good solution enabling small training loss, and will suffer from poor validation performance. The right plot in Fig. 10 in the revised paper illustrates how different algorithms perform in the validation dataset. It is observed that the smaller training loss the algorithm reaches (i..e, the faster the algorithm converges), the better the final validation performance it achieves.
>
> **Q6 [Neural network architecture].** We utilize ResNet-20 on CIFAR-10 and ResNet-50 on ImageNet. We have added these settings in the revised paper.

---

### Author Response · Authors · 2021-11-22
**Revision Summary**

Dear Reviewers,

Many thanks for your careful review and valuable feedback. We have attempted to address all comments as best as we can. Following your feedback, we have made several revisions on the draft, which are highlighted in **Blue** in the new submission. Here is a summary of all finished revisions.

1. In response to Reviewers 1RX3 and RKoY's comments, we have established lower bound in Proposition 1 and 2.
2. In response to Reviewer 1RX3's comments, we have added Fig. 10 to illustrate how the data heterogeneity will influence both the training loss and validation accuracy, and Fig. 6 to illustrate how different algorithms (including the reparameterized DAdam proposed by the reviewer) perform in terms of loss.
3. In response to Reviewer RKoY's comments, we have added Section D (as well as Figs. 7 and 8) in the appendix to discuss the influence of hyper-parameter $\nu$ on DSGD, DAdam, and DAG-Adam.
4. We corrected typos in Algorithms 1, 2, and 3.  These typos do not influence the derived convergence analysis (because the analysis is conducted based on the correct form (20) - (23)). We are very grateful to  Reviewer XYfg for pointing it out.
5. Other revisions include more detailed experimental setting description, replacing $L(x^\star)$ with $L^\star$ in Theorem 1, adding standard error to each item in Tables 1 and 2, and other slight modification in response to reviewers' minor comments.

Best Regards,

Authors

---

### Decision · Program_Chairs · 2022-01-20

**Decision:**

Reject

**Comment:**

The paper proposes a ''communicate-then-adapt'' framework for decentralized optimization, with both theoretical and empirical analysis. The reviewers' main concern is the comparison in theory with prior methods like the GT-DAdam. The convergence to a stationary point of GT-DAdam seems to be faster than the proposed method in the important non-convex optimization. The reviewers are not convinced by the strong claim that ''communicate-then-adapt'' is better than ''adapt-then-communicate'' as such ''adapt-then-communicate'' method can also achieve same or better rates, possibly with less hyper-parameter tuning. I would suggest the authors to make more proper comparison with related methods.